# Sign-In to the Lottery:
# Reparameterized Sparse Training

**Advait Gadhikar**[*]   **Tom Jacobs**[*]   **Chao Zhou**   **Rebekka Burkholz**
CISPA Helmholtz Center for Information Security, Saarbrücken, Germany
{advait.gadhikar, tom.jacobs, chao.zhou, burkholz}@cispa.de

## Abstract

The performance gap between training sparse neural networks from scratch (PaI) and dense-to-sparse training, presents a major roadblock for efficient deep learning. According to the Lottery Ticket Hypothesis, PaI hinges on finding a problem specific parameter initialization, given a sparse mask. As we show, to this end, determining correct parameter signs is sufficient. Yet, they remain elusive to PaI. To address this issue, we propose Sign-In, which employs a dynamic reparameterization that provably induces sign flips. Such sign flips are complementary to the ones that dense-to-sparse training can accomplish, rendering Sign-In as an orthogonal method. While our experiments and theory suggest performance improvements of PaI, they also carve out the main open challenge to close the gap between PaI and dense-to-sparse training.

## 1   Introduction

As neural network sizes scale to billions of parameters, building efficient training pipelines is of paramount importance. Sparsification plays a pivotal role in enabling such efficient pipelines (Frantar & Alistarh, 2023). State-of-the-art sparsification methods rely on eliminating parameters during training (Kuznedelev et al., 2024). The success of these methods can be attributed to finding an optimal combination of parameters and sparse masks (Paul et al., 2023). However, identifying them requires training a dense network before it can be sparsified such that information from the dense network can be inherited by the subsequent sparse network (Gadhikar & Burkholz, 2024).

It is then no surprise that the most successful sparsification methods rely on dense training phases. Dense-to-sparse methods like gradual pruning (Han et al., 2015) or continuous sparsification (Kusupati et al., 2020) explicitly start from a dense network. However, this comes at the cost of high computational and memory demands during the dense training phase. Moreover, the benefits of sparsification become mostly available towards the end of training or during inference.

Is there a way to train sparse networks from the start, given a sparse mask? The Lottery Ticket (LT) Hypothesis (Frankle & Carbin, 2019) posits that there exist sparse networks at random initialization which enjoy optimal parameter-mask coupling and can achieve the same performance as their dense counterpart. Finding such sparse networks at initialization would unlock the benefits of sparse training from scratch. A plethora of pruning at initialization (PaI) methods have attempted to find such sparse networks at initialization (Lee et al., 2019; Tanaka et al., 2020; Wang et al., 2020), however, they struggle to match the generalization performance of dense-to-sparse training methods.

Understanding and overcoming the limitations of PaI remains an open research problem and is the focus of our work. The main challenge for PaI methods is to find highly performant parameter-mask combinations by leveraging task-specific information efficiently (Kumar et al., 2024; Frankle et al., 2021). The problem is thus comprised of two parts: a) learning a mask and b) optimizing the

---

[*]Equal contribution

39th Conference on Neural Information Processing Systems (NeurIPS 2025).

weights corresponding to the mask, which is hard if the mask is extremely sparse. Here we are less concerned about learning a mask, but instead focus on the hard problem of optimizing parameter-mask combinations, given a mask at initialization. At the center of our investigation is the question:

*What hinders training a sparse mask, and can the gap between PaI and dense-to-sparse be closed?*

The key to addressing this question lies in promoting parameter *sign flips*. Echoing the analysis of learning rate rewinding (LRR) by Gadhikar & Burkholz (2024), through a broader empirical investigation of dense-to-sparse training methods, we identify a critical phenomenon responsible for finding optimal parameter-mask combinations: the ability to flip parameter signs during training. We term this phenomenon sign alignment, as the signs encode all relevant information additionally to the mask (and data) to enable effective training from scratch. Moreover, as we show, sign alignment occurs early in the dense-to-sparse training process, promoting convergence toward flatter minima. In contrast, PaI faces challenges in stabilizing sign flips. In line with this observation, it was shown in a theoretically tractable setting of one-hidden-neuron that training a sparse network from scratch fails to flip signs from a bad initialization (Gadhikar & Burkholz, 2024).

Motivated by this insight, we propose a way to flip and learn correct signs during sparse training from scratch in a higher number of cases (see Figure 2). We achieve this via *Sign-In*, a reparameterization $m \odot w$ of the non-masked parameters, providing them with an inner degree of freedom (where $m$ and $w$ are continuous parameters). Note that this reparameterization, depending on the inner scaling, has a bias towards sparsity. This has been used by the continuous sparsification methods Spred (Ziyin & Wang, 2023) and PILoT (Jacobs & Burkholz, 2025). In contrast, we do not use the reparameterization for sparsification but for improving the optimization geometry to enable sign alignment and thus seek to counteract its sparsity bias. We propose an inner scaling that promotes sign flips by rebalancing network layers and, correspondingly, the speed in which they learn.

This provably recovers a different sign case than dense training and could thus be seen as complementary (see Figure 2). Moreover, in combination with a moderate degree of over-parameterization, which tends to be common in PaI, *Sign-In* fully solves the one-hidden-neuron problem. In line with this insight, we verify in experiments that existing dense-to-sparse methods can be improved by utilizing *Sign-In*. Most importantly, we demonstrate across multiple experiments that *Sign-In* significantly enhances the performance of sparse training from scratch, including those using random masks. We attribute this improvement to better sign alignment and the ability to discover flatter minima. However, as we show, *Sign-In* cannot replicate the same sign flipping mechanism as that of initial dense training, and is still not competitive with the accuracy of dense-to-sparse methods.

This poses the question: Could there exist another unknown parameterization that could replace the sign flipping mechanism of dense training? Unfortunately, the answer is negative, as our proof of an impossibility statement reveals. While we can report progress in the right direction, it remains an open problem to completely close the gap between PaI and dense-to-sparse methods.

**Contributions** 1) We empirically show that dense-to-sparse training methods identify trainable parameter-mask combinations via early sign alignment allowing them to find flatter minima and achieve state-of-the-art performance. 2) We provide evidence for the following hypothesis: Across dense-to-sparse training masks, the learned signs define an initialization that enables competitive sparse training from scratch. A fully random initialization (that is decoupled from the mask) performs on par or worse than PaI (including random pruning). 3) To learn signs more effectively, we propose *Sign-In*, a weight reparameterization, which can provably —using Riemannian gradient flow and dynamical systems analysis —recover correct signs in a case that is complementary to overparameterization. 4) Extensive empirical evaluation shows that *Sign-In* can significantly improve the performance of sparse training from scratch by improving sign alignment and finding flatter minima.

## 2  Related Work

**Lottery Ticket Hypothesis (LTH)** Training sparse neural networks from scratch is a hard problem (Kumar et al., 2024), but Frankle & Carbin (2019) posit that there exist initializations that make sparse networks trainable, i.e., Lottery Tickets. However, their approach to find such initializations by iterative magnitude pruning (IMP) is computationally expensive and its performance does not transfer to larger models, even though lottery tickets should exist in theory da Cunha et al. (2022); Burkholz

(2022a,b); Burkholz et al. (2022); Burkholz (2024). Renda et al. (2020) improved its accuracy by Weight Rewinding (WR) and Learning Rate Rewinding (LRR). Both WR and LRR are similarly expensive, but inherit the initialization from denser networks.

**Dense-to-sparse methods** While dense training has been conjectured to be a critical factor to enter a good loss basin Paul et al. (2023) or quadrant Gadhikar & Burkholz (2024), other methods aim to reduce its duration. Gradual pruning methods like AC/DC (Peste et al., 2021), CAP (Kuznedelev et al., 2024), and WoodFisher (Singh & Alistarh, 2020) prune based on a parameter importance score, interspersed with training. Continuous sparsification methods such as STR (Kusupati et al., 2020), spred (Ziyin & Wang, 2023), and PILoT (Jacobs & Burkholz, 2025) reparameterize a dense neural network to induce sparsity. Although such dense training phases seem to be critical to learn good parameter signs Gadhikar & Burkholz (2024), we propose a complementary method, which also works for pruning at initialization.

An orthogonal class of algorithms, dynamic sparse training, includes RiGL (Evci et al., 2020; Lasby et al., 2023; Chen et al., 2021), SET (Mocanu et al., 2018), MEST (Yuan et al., 2021), CHTs (Zhang et al., 2025) which dynamically modify a sparse mask during training. However, our work focuses on training a fixed mask from scratch.

**Pruning at initialization (PaI)** A variety of methods have been proposed to sparsify networks at initialization based on different scores that assess parameter importance like saliency in SNIP (Lee et al., 2019) and GraSP (Wang et al., 2020), signal flow in Synflow (Tanaka et al., 2020) and maximal paths (Pham et al., 2023). Other methods include, DPaI (Xiang et al., 2025), Phew(Patil & Dovrolis, 2021), ProsPr (Alizadeh et al.). However, randomly pruned sparse network can perform at par with these methods (Liu et al., 2021; Gadhikar et al., 2023; Ma et al., 2021; Su et al., 2020), calling our ability to identify problem specific masks into question (Frankle et al., 2021; Jain et al., 2024). Kumar et al. (2024) characterize the challenge from an information-theoretic lens, showing that initial dense training, in contrast to PaI, increases effective model capacity. Our goal is not to address the mask identification problem but to improve sparse training given a mask, by a reparameterization.

**Reparameterization** Reparameterizations have been studied from the perspective of implicit bias and gradient flow (Chizat & Bach, 2020; Li et al., 2022; Woodworth et al., 2020; Gunasekar et al., 2020, 2017; Chou et al., 2024; Vaškevičius et al., 2019; Li et al., 2023; Zhao et al., 2022; Li et al., 2021; Dominé et al., 2024; Jacobs & Burkholz, 2025; Jacobs et al., 2025; Marcotte et al., 2023, 2024; Liang & Montufar, 2025). It explains how overparameterization can improve generalization and how this is affected by various training factors such as the initialization, learning rate, noise, and explicit regularization. An important assumption in these works is that the activation function is linear. This simplification enables the reparameterization to be expressed using mirror flow or Riemannian gradient flow frameworks (Li et al., 2022). In contrast, we employ a reparameterization specifically to induce sign flips in the presence of a non-linear activation function. Our proposed reparameterization has been applied to induce continuous sparsification (Ziyin & Wang, 2023; Jacobs & Burkholz, 2025; Kolb et al., 2025) with the help of explicit regularization. To combat this trend, we rescale the parameters dynamically exploiting a rescale invariance.

## 3 Correct Signs Facilitate Sparse Training

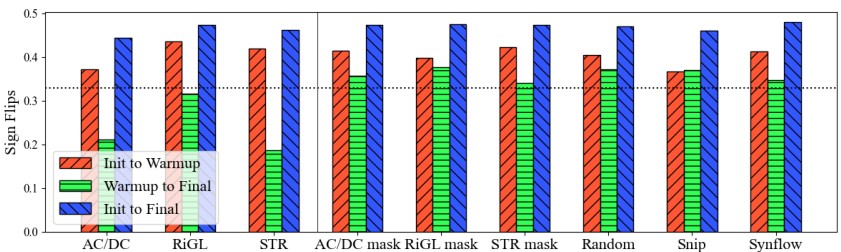

Figure 1: **Sign learning** a 90% sparse ResNet50 on ImageNet. The majority of signs are flipped early for dense-to-sparse methods, upto warmup. Moreover, signs stabilize after warmp-up. In contrast, signs do not stabilize after warmup for sparse training from scratch for different masks.

Paul et al. (2023) have established that iterative pruning methods like Learning Rate Rewinding (LRR) and Weight Rewinding (WR) rely on the effective exploration of the loss landscape during dense overparameterized training. Dense-to-sparse training inherits its performance by training overparameterized in the beginning, identifying a good loss basin. Gadhikar & Burkholz (2024) have shown that this loss basin can be characterized by the weight signs and thus the quadrant of the solution. Our first objective is to show that this finding is not unique to LRR. In fact, it is a common phenomenon that points to a critical limitation of PaI to learn signs.

Our experiments reveal two key findings: (i) Initializing a network with an aligned sign and mask is sufficient for successful sparse training. (ii) Sign alignment happens early during training for dense-to-sparse methods and guides the training trajectory towards flatter minima. Specifically, we study three well-established sparsification methods to draw insights on sign alignment: magnitude based pruning with AC/DC (Peste et al., 2021), continuous sparsification with STR (Kusupati et al., 2020), and dynamic sparse training with RiGL (Evci et al., 2020). The first two methods leverage information from dense training by alternating dense and sparse training and gradual sparsification by soft thresholding respectively. Although, RiGL does not involve a dense training phase, it retains dense gradient information, which we observe serves a similar purpose. For comparison with sparse training, we train a randomly pruned sparse mask from scratch, which has been shown to be competitive with (and even better than) other PaI methods (Liu et al., 2021) and serves as a strong baseline. Noteworthy is the fact that the random mask contains no information about the task. Even though other masks should therefore have an advantage, as we will see later, they cannot shine through because of limited trainability.

**Signs are learned early** To study the alignment between the sparse mask and parameter signs, we track them during dense-to-sparse training of a ResNet50 on ImageNet (see Section 6 for details). As shown in Figure 1, the initial signs and final learned signs are significantly different since approximately 50% of the signs are flipped for all methods (blue bar), i.e., the signs at initialization have a close to random overlap with the final learned signs. As training progresses, upto the warmup phase (10-th epoch), a significant number of signs flip, as shown by the red bar in Figure 1. After the warmup phase, fewer signs flip in addition (green bar in Figure 1). In comparison, training the same masks from scratch starting from a random re-

Table 1: **Sign alignment** for a ResNet50 trained on ImageNet with 90% sparsity. Baseline performance is recovered when the learned mask is initialized with the learned parameter signs, but not with the learned parameter magnitude and random signs.

| Method | AC/DC | STR | RiGL |
|---|---|---|---|
| Baseline | 74.68 | 73.65 | 73.75 |
| Epoch 10 sign $_{+\text{ learned mask}}$ | 73.97 | 71.7 | 73.32 |
| Epoch 30 sign $_{+\text{ learned mask}}$ | **74.89** | **73.91** | 73.7 |
| Final sign $_{+\text{ learned mask}}$ | 74.88 | 73.77 | **73.74** |
| Epoch 10 mag $_{+\text{ learned mask}}$ | 70.93 | 68.01 | 72.30 |
| Final mag $_{+\text{ learned mask}}$ | 70.94 | 68.35 | 72.40 |
| Random init $_{+\text{ learned mask}}$ | 70.6 | 68.38 | 71.89 |

initialization (which is decoupled from the dense-to-sparse training procedure) does not identify stable signs early. The performance of these masks is at par with a random mask trained from scratch (see Table 8 in the appendix) in the absence of sign alignment. Moreover, the sign flip statistics show a similar pattern as training a random mask from scratch. A significant number of sign flips occur after warmup, and the overlap between initialization, warmup and the final signs is closer to random (see Figure 1). Across all three dense-to-sparse baselines, we observe that signs are learned early during training and remain largely stable after warmup. In contrast, sparse training does not find stable signs to enable sign alignment, reflecting in its worse performance.

**Signs are more informative than magnitude** As we verify next, the critical information about the task that is missing from the mask are the parameter signs. To do so, we use signs learned at different stages during training, the learned mask and random magnitudes for each method to initialize a sparse network. As reported in Table 1, final signs combined with the mask are sufficient for sparse training and achieve performance comparable to the baseline. Moreover, since the signs are identified as early as the 10-th epoch during training, these early signs (combined with the mask) recover baseline performance with an error of $\approx 1\%$. Full baseline performance can be recovered by using signs from the 30-th epoch. In contrast, if we initialize the sparse network with magnitudes learned at

intermediate epochs, learned mask and random signs, we barely improve over training a randomly intialized sparse mask from scratch Table 1.

Our results not only suggest that sign alignment is sufficient for sparse training, but it also happens early during dense-to-sparse training. Sparse training from scratch with falls short in comparison.

**Flatter loss landscapes with dense-to-sparse methods** To dive deeper into the benefits of sign alignment for trainability, we track the maximum eigenvalue of the Hessian of the loss as a proxy of sharpness (Jiang et al., 2019). It can also be interpreted as a proxy measure for the tractability (or conditioning) of the optimization problem. As shown in Figure 4(b), the dense-to-sparse baselines AC/DC, STR and RiGL have smaller sharpness indicating a flatter loss landscape once signs align during training. In comparison, the random sparse network converges to a sharper minimum.

Our observations lead to the conclusion that dense-to-sparse methods are effective because they inherit critical information from dense training in the form of *parameter signs* during the early training phase. These signs play a crucial role in dictating the training trajectory of the sparse network. In contrast, this mechanism is absent in training a fixed mask from scratch, resulting in poor performance and a sharper loss landscape during training, which indicates convergence to an inferior loss basin.

A natural question arises: Can we achieve a similar sign alignment in sparse training from scratch? While it seems impossible to replicate the benefits of initial dense training while starting sparse, we demonstrate that sign alignment can be improved using an orthogonal sign flipping mechanism, which we induce by a network reparameterization. As we show, this reparameterization helps to learn correct ground truth signs in a theoretically tractable case of a single neuron two-layer network.

## 4   Sign-In: Learning Sign Flips

To improve the learning of task relevant sign flips, we introduce a reparameterization. Every weight parameter $\theta$ is replaced by the product of two parameters $m \odot w$, where $\odot$ denotes a pointwise multiplication (Hadamard product). This introduces an additional degree of freedom $\beta$, which is determined by the initialization and captured by $m^2 - w^2 = \beta \mathbf{1}$, where $\beta > 0$ and $\mathbf{1} = (1, ..., 1)$. According to Lemma 4.8 (Li et al., 2022), the reparameterization induces a Riemannian gradient flow in the original parameter $\theta$:

$$d\theta_t = -\sqrt{\theta_t^2 + \beta} \odot \nabla L(\theta)dt, \qquad \theta_0 = \theta_{init} \tag{1}$$

This holds for general continuously differentiable objective function $L : \mathbb{R}^m \to \mathbb{R}$. To induce helpful sign flips, we have to deal with two factors: stochastic noise (Pesme et al., 2021) and weight decay (Jacobs & Burkholz, 2025). Both factors have the same effect on the training dynamics: They induce sparsity by shrinking $\beta$. Yet, if $\beta = 0$, we cannot sign flip due to the $\sqrt{\theta_t^2}$ in the Riemannian gradient flow Eq. (1). We address this issue with Algorithm 1 in Appendix B by resetting the inner scaling $\beta$ without changing the weights of the network every $p$ epochs up to epoch $T_2$. In all our experiments, we choose the same scaling of $\beta = 1$ for all parameters, as this leads to stable results. We employ an analytic solution to apply the rescaling on both $m$ and $w$, as given in Appendix C. This gives the weights more opportunities to align their signs correctly, as proven for a simple example in Section 5.

**Computational and memory cost** Sign-In doubles the amount of parameters relative to sparse training. This is also the case for continuous sparsification methods that utilize the implicit bias of $m \odot w$ towards sparsity, like spred (Ziyin & Wang, 2023) and PILoT (Jacobs & Burkholz, 2025). Nevertheless, according to Ziyin & Wang (2023), the training time of a ResNet50 with $m \odot w$ parameterization on ImageNet increases roughly by 5% only and the memory cost is negligible if the batch size is larger than 50. Furthermore, at inference, we would return to the representation $\theta$ by merging $\theta = m \odot w$. Therefore, *Sign-In* is able to improve sparse training with minimal overhead as also highlighted by a FLOP comparison in Table 15.

## 5   Sign Learning in Theory

Gadhikar & Burkholz (2024) proposed a simple testbed to compare two iterative pruning algorithms, IMP and LRR. LRR's superior performance is explained by its ability to flip the sign of a critical parameter during a dense training phase. As we show, the same mechanism distinguishes sparse

training from scratch with random initialization and dense training. The next section is dedicated to the question how *Sign-In* changes this picture. Remarkably, it provably resolves another case than dense (overparameterized) training. In combination with dense training, it can flip even all relevant signs and solve the learning problem (see Figure 2).

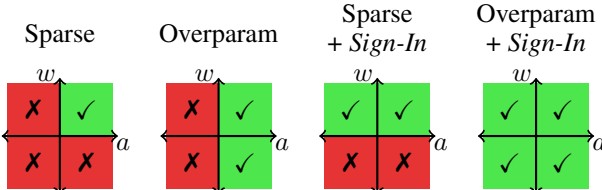

Figure 2: ***Sign-In* recovers the solution in a different case than overpameterization** for a two-layer network by sign flipping. Combining the two solves all cases (empirically).

**Single neuron setting** Let $\{\mathbf{z}_i\}_{i=1}^n$ be a data set of i.i.d $\mathcal{N}(\mathbf{0}, \mathbf{I})$ random variables. Consider a two-layer neural network consisting of a single hidden neuron $f : \mathbb{R}^d \to \mathbb{R}$ such that $f(\mathbf{z} \mid a, \mathbf{w}) := a\sigma\left(\mathbf{w}^T \mathbf{z}\right)$, where $\sigma(\cdot) := \max\{\cdot, 0\}$ denotes the Rectified Linear Unit (ReLU). The labels $\{y_i\}_{i=1}^n$ are generated from a sparse teacher $y_i = \tilde{f}\left(\mathbf{z}_i \mid \tilde{a}, \tilde{\mathbf{w}}\right)$ with $\tilde{a} > 0$ and $\tilde{\mathbf{w}} = (1/\tilde{a}, 0, 0, .., 0)$. On this data, we train a student two-layer neural network with gradient flow and randomly balanced parameter initialization, i.e., $a_{\text{in}}^2 = \|\mathbf{w}_{\text{in}}\|_2^2$. The objective is to minimize the mean squared error (MSE), which is given by $L(a, \mathbf{w}) := \frac{1}{2n} \sum_{i=1}^n \left(f\left(\mathbf{z}_i \mid a, \mathbf{w}\right) - y_i\right)^2$. The training dynamics with loss function $L$ is described by the gradient flow:

$$\begin{cases} da_t = -\partial_a L(a_t, \mathbf{w}_t)dt, & a_0 = a_{\text{in}} \\ d\mathbf{w}_t = -\nabla_{\mathbf{w}} L(a_t, \mathbf{w}_t)dt, & \mathbf{w}_0 = \mathbf{w}_{\text{in}}. \end{cases} \tag{2}$$

In this setting, we are interested in comparing two cases: a) We call the multi-dimensional case with $d > 1$ the overparameterized case, as we could also set the weight parameters $w_j$ with index $j > 1$ to zero. We also denote the student as $f_1$. Dense-to-sparse training would typically first train all parameters, then prune all weights with $j > 1$, and continue training only $w_1$ and $a$ after pruning. b) The one-dimensional case $d = 1$ can be seen as special case where we mask the components $j > 1$ from the start. It is equivalent to considering a 1-dimensional problem with $d = 1$. We also write $f_0$ for the student, which faces generally a harder problem, as we establish next.

**Sparse training** If the student is one-dimensional ($d = 1$), it can only recover the ground truth if it is initialized with the correct signs, i.e. when $a_{in} > 0$ and $w_{in} > 0$.

**Theorem 5.1.** *(Theorem 2.1 (Gadhikar & Burkholz, 2024)) In the one-dimensional single neuron setting, Eq. (2) with $d = 1$, the student can recover the ground truth given sufficiently many samples, if $a_{in} > 0$ and $w_{1,in} > 0$. In all other cases ($a_{in} > 0$, $w_{1,in} < 0$), ($a_{in} < 0$, $w_{in} > 0$) and ($a_{in} < 0$, $w_{1,in} < 0$), it will not attain the correct solution.*

***Sign-In* reparameterization** The *Sign-In* reparameterization, which we have introduced in Section 4, can resolve the additional case $a_{in} < 0, w_{1,in} > 0$ when $d = 1$. Note that both $a$ and $w$ are reparameterized with different inner scalings denoted by $\beta_1$ and $\beta_2$. The corresponding Riemannian gradient flow in the original parameters $a$ and $w_1$ is:

$$\begin{cases} da_t = -\sqrt{a_t^2 + \beta_1}\,\partial_a L(a_t, w_{1,t})dt, & a_0 = a_{\text{in}} \\ dw_{1,t} = -\sqrt{w_{1,t}^2 + \beta_2}\,\partial_{w_1} L(a_t, w_{1,t})dt, & w_{1,0} = w_{1,\text{in}}. \end{cases} \tag{3}$$

If $w_{1,t} > 0$ for all $t > 0$, $L$ is continuously differentiable.

**Theorem 5.2** (Sign-In improves PaI). *Let $a$ and $w_1$ follow the* Sign-In *gradient flow dynamics in Eq. (3) and let $\beta_1 > \beta_2 > 0$. If $w_{1,in} > 0$, then the student $f_0$ can learn the correct target with probability $1 - \left(\frac{1}{2}\right)^n$. If $w_{1,in} \le 0$, it still fails to recover the ground truth.*

Proof. The proof is given after Theorem A.1 in the appendix. The main steps are: a) Characterization of the gradient field at balanced initialization, breaking the coupling that causes it to approach zero. b) Determining the direction of the stable and unstable manifold at the origin (saddle point). c) Demonstrating that the dynamical system must converge towards a stationary point.

*Remark* 5.3. The proof combines tools from dynamical systems theory and mirror flow theory to show convergence, which might be of independent interest.

To illustrate Theorem 5.2, we simulate the toy setup for both the standard gradient flow and the reparameterized gradient flow. Figure 3(a) shows that the standard gradient flow converges only when the signs are correctly initialized, whereas the reparameterized model also succeeds if $a_{in} < 0, w_{1,in} > 0$. Other initializations lead the student to converge to the origin and thus a function that is constant at 0, i.e. training fails.

Note that the condition $\beta_1 > \beta_2 > 0$ in Theorem 5.2 is necessary. In practice, this suggests that each layer would require different scaling, which can be hard to tune. We show next that this is not needed and the same scaling can be used for each layer in practice.

**Multi-dimensional input** For multi-dimensional input ($d > 1$), the outer layer parameter $a$ cannot sign flip like in case of $d = 1$ (Lemma 2.2 (Gadhikar & Burkholz, 2024)). However, we have shown that when the rescaling of $a$ is larger then that of $w_1$, i.e., $\beta_1 > \beta_2 > 0$, the sign of $a$ is recovered, if $w_1$ initialized with the correct sign. This is impractical for larger neural networks with more layers, since we have to choose a different scaling $\beta$ for all layers. Nonetheless, for a multi-dimensional input layer, different rescaling is not necessary, as we verify empirically next.

We train the student network with $d = 5$ for time $T = 200$ with gradient descent and a learning rate $\eta > 0$ such that total steps are $T/\eta$. The experiment is repeated for 100 different initializations. The success percentages training runs are presented in Table 2. We observe that *Sign-In* consistently recovers the sign of $a$, with same scaling $\beta = 1$.

A potential explanation is that a finite learning rate $\eta > 0$ induces overshooting, leading to sign flips. However, gradient flow with the standard parameterization does not recover the correct sign of $a$ due to this overshooting. Accordingly, the success of *Sign-In* cannot be attributed entirely to this factor. The key factor is that the balance between $a$ and $||\mathbf{w}||_{L_2}$ is broken. Intuitively, for $a_t$ to sign flip, it needs to move faster than $||\mathbf{w}_t||_{L_2}$ as in the one-dimensional

Table 2: Percentage of sign recoveries if $a_0 < 0$ for standard and *Sign-In* gradient descent using different learning rates $\eta$.

| Method | GF | *Sign-In* GF |
|---|---|---|
| $\eta = 0.001$ | 0% | 100% |
| $\eta = 0.01$ | 0% | 100% |

case (see Figure 3(a)). We observe that $|a_{in}|^2 = ||\mathbf{w}_{in}||_{L_2}^2 \geq w_{k,in}^2$ for all $k \in [d]$. Therefore, the reparameterization induces a relative speed up for $a$ compared to $||\mathbf{w}||_{L_2}$, i.e.,

$$\sqrt{a_{in}^2 + \beta} = \sqrt{||\mathbf{w}_{in}||_{L_2}^2 + \beta} \geq \frac{1}{n} \sum_{k=1}^{n} \sqrt{w_{k,in}^2 + \beta} \tag{4}$$

This indicates $a$ can potentially sign flip without different rescaling in-between layers.

**No replacement for overparameterization** While it is remarkable that the combination of *Sign-In* and overparameterized training can solve our theoretical toy problem, ideally, we would like to facilitate training from scratch ($d = 1$) without utilizing any dense training. *Sign-In* is not able to achieve this without dense training but does there exist another reparameterization that could? Unfortunately, the answer is negative, as we establish next.

Consider the one-dimensional problem again, with a continuous differentiable reparameterization $g_1 \in C^1(M_1, \mathbb{R})$ and $g_2 \in C^1(M_2, \mathbb{R})$ where $M_1$ and $M_2$ are smooth manifolds. Then we reparametrize both $a$ and $w$ such that $a = g_1(\mathbf{a})$ and $w_1 = g_2(\mathbf{w})$.

**Theorem 5.4** (No reparameterization replaces overparameterization). *If $w_{1,in} < 0$, then there exists no continuous parameterization $(g_1, g_2)$ for which the resulting gradient flow reaches the global optimum.*

Proof. The flow has to move through $w_1 = 0$. We can assume there exists an $\mathbf{w}^* \in M_2$ such that $g(\mathbf{w}^*) = 0$, otherwise it is impossible to move through $w_1 = 0$. This implies that $\partial_a L = \partial_{w_1} L = 0$. Thus, no continuously differentiable reparameterization can recover the sign of $w_1$. □

According to Theorem 5.4, from an optimization perspective, we can not replace overameterization with a reparameterization to recover the second case in Figure 2. This impossibility further emphasizes the difficulty of optimizing sparse networks without a dense training phase.

**Multiple neurons** Do our insights still hold if we increase the number of neurons? To verify the broader validity of our claims, we consider a neural network $f_2 : \mathbb{R}^d \to \mathbb{R}$ given by $f_2(\mathbf{z} \mid \mathbf{a}, \mathbf{W}) := \sum_{i=1}^{k} a_i \sigma\left(\mathbf{w}_i^T z\right)$. The data is generated from a similarly structured teacher network $\tilde{f}_2$ with $k = 3$ neurons and input dimension $d = 2$. The weights are initialized and trained as outlined in Appendix D. A difficult sign initialization is chosen as $a_i > 0$ for all $i \in [3]$. Figure 3 illustrates each neuron by a vector $|a_i|\mathbf{w}_i$. The teacher neurons are vectors touching the unit circle and student neurons are denoted with markers.

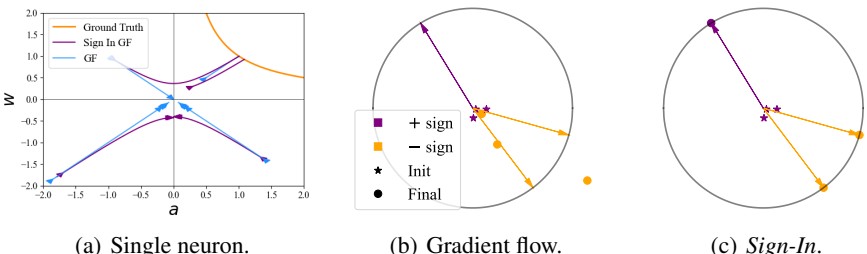

(a) Single neuron.    (b) Gradient flow.    (c) *Sign-In*.

Figure 3: **Student-teacher setup** (a) Training **one neuron** for a single input $d = 1$, when $w_1 \leq 0$ both methods fail to reach the ground truth. If $w_1 > 0$, the *Sign-In* gradient flow succeeds, whereas gradient flow fails in this case when additionally $a < 0$. (b) Representation of a two layer student-teacher neural network with **multiple neurons** inspired by Chizat et al. (2020) for sparse training from scratch fails with bad signs. (c) *Sign-In* enables learning the representation with bad signs.

## 6 Experiments

We show that *Sign-In* provably improves PaI in a small scale example (Theorem 5.2), now we have to verify that its merits also translate to larger scale settings. While our main focus is on PaI and thus training sparse networks from scratch, our theory implies that *Sign-In* could also boost dense-to-sparse training performance, as their combination can resolve cases where either would fail. However, we do not expect training from scratch with *Sign-In* to outperform training starting from well aligned signs, as *Sign-In* acts orthogonally to dense-to-sparse training in our theoretical setting. *Sign-In* is expected to facilitate different sign flips.

**Experimental Setup** We perform experiments on standard vision benchmarks including CIFAR10, CIFAR100 (Krizhevsky et al., 2009), and ImageNet (Deng et al., 2009) training a ResNet20, ResNet18, and ResNet50 model, respectively. For architectural variation, we also train a vision transformer DeiT Small (Touvron et al., 2021) on ImageNet. All training details are provided in Table 4 in Appendix E.

Table 3: Sparse training with *Sign-In*.

| Dataset | Method | $s = 0.8$ | $s = 0.9$ | $s = 0.95$ |
|---|---|---|---|---|
| CIFAR10 $_{+\text{ResNet20}}$ | Random | $88.25(\pm0.35)$ | $86.25(\pm0.3)$ | $83.56(\pm0.23)$ |
| | Random + *Sign-In* | $\mathbf{89.37(\pm0.14)}$ | $\mathbf{87.83(\pm0.11)}$ | $\mathbf{84.74(\pm0.06)}$ |
| CIFAR100 $_{+\text{ResNet18}}$ | Random | $73.95(\pm0.1)$ | $72.96(\pm0.24)$ | $71.36(\pm0.16)$ |
| | Random + *Sign-In* | $\mathbf{75.32(\pm0.25)}$ | $\mathbf{73.94(\pm0.13)}$ | $\mathbf{72.51(\pm0.14)}$ |
| ImageNet $_{+\text{ResNet50}}$ | Random | $73.87(\pm0.06)$ | $71.56(\pm0.03)$ | $68.72(\pm0.05)$ |
| | Random + *Sign-In* | $\mathbf{74.12(\pm0.09)}$ | $\mathbf{72.19(\pm0.18)}$ | $\mathbf{69.38(\pm0.1)}$ |

**Scaling reset** An integral part of *Sign-In* is that we dynamically reset the internal scaling to combat the implicit bias towards sparsity of the parameterization. To ablate its contribution, we report results for *Sign-In* with and without resetting the scaling for a random mask in Table 14 in the appendix. Indeed, without the scaling reset, the performance gain is less significant. This confirms that resetting mitigates the effect of stochastic noise and weight decay, enabling better sign flips.

*Sign-In* **improves sparse training** Table 3 reports results for sparse training with a random mask. The layerwise sparsity ratios are chosen as balanced such that each layer has the same number of nonzero

parameters (Gadhikar et al., 2023). Sparse training with *Sign-In* as per Algorithm 1 consistently outperforms standard training, substantiating our theoretical results. A similar improvement is observed for other PaI methods in Table 16 in Appendix H. We apply *Sign-In* also to different sparse masks identified with PaI or dense-to-sparse methods as reported in Table 8 in Appendix H. While layerwise sparsity ratios determine the overall trainability of a sparse network, *Sign-In* improves performance in better trainable settings.

**Vision transformer with *Sign-In*** *Sign-In* is also able to improve sparse training of a randomly masked vision transformer. A DeiT Small architecture (trained for 300 epochs from scratch) gains more than $1\%$ in performance with *Sign-In*, as reported in Table 10, Appendix J.

**Improved sign alignment** Figure 4(a) validates the *Sign-In* mechanism by tracking the percentage of sign flips during training for a $90\%$ sparse ResNet50 on ImageNet. Across the full duration of training, *Sign-In* enables strictly more sign flips than sparse training. This improves sign alignment, as reflected in the improved performance and the flatter minima (see Figure 4(b)). Since *Sign-In* employs an alternate mechanism for sign flipping, it cannot learn stable signs early like dense-to-sparse methods. Instead, it enables more sign flips throughout training as compared to standard sparse training, to aid alignment by better exploration of the parameter space.

***Sign-In* improves other sparse training methods** Only a combination of *Sign-In* and overparameterization is able to solve all sign flip cases in our theoretical setting, as also confirmed by numerical experiments in Table 6, Appendix F. Therefore, we expect the combination of dense-to-sparse or dynamic sparse training methods with *Sign-In* to achieve further gains in performance. Appendix K shows that both AC/DC and RiGL are improved with *Sign-In*.

**Implications of theory** The surprising success of *Sign-In* combined with overparameterization in our theoretical setting relies on the last layer $a$ learning at a higher speed than the first layer $w_1$. This enables $a$ to flip its sign if necessary, which is impossible in the same setting without *Sign-In*. $w_1$ can follow once the sign of $a$ is correctly learned. We conjecture that weights in layers close to the input would have a harder time flipping in deep neural networks consisting of more layers and neurons, as other layers on top slow its learning down. In line with this narrative, *Sign-In* promotes more sign flips in layers closer to the output, as shown in Figure 7 in the appendix.

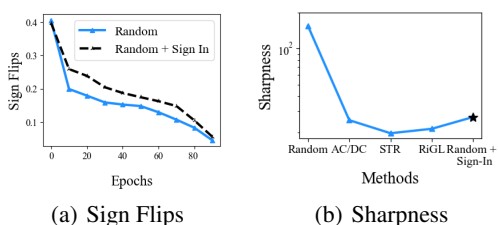

(a) Sign Flips          (b) Sharpness

Figure 4: **Sign flips with *Sign-In*** Training a random, $90\%$ sparse ResNet50 on ImageNet with *Sign-In* induces (a) more sign flips during training and finds (b) a flatter minimum.

To understand how these sign flips relate to generalization, we must analyze more complex scenarios in future and see whether more effective sign learning could further boost dense-to-sparse training. The primary focus of this work has been to improve sparse training from scratch and *Sign-In* overcomes one of its major limitations regarding learning parameter signs.

## 7 Conclusion

Parameter signs that are aligned with a given mask are sufficient to train a sparse network from scratch. We have empirically verified this conjecture for masks that result from dense-to-sparse training algorithms. Regardless of the sparse mask, learning highly performant parameter signs is a great challenge for PaI. To aid parameter sign flipping, we have proposed to reparameterize a neural network by *Sign-In*. For a single neuron network, we have proven with theory from Riemannian gradient flows and dynamical systems that *Sign-In* resolves a case that is complementary to overparameterized training which translates to largescale performance. Ideally, we would like to make dense training phases obsolete to save computational resources. Although, as we have shown, a reparameterization alone will not be able to achieve this. Closing the gap between training sparse, randomly initialized networks from scratch and dense-to-sparse training remains a hard open problem. Our work brings us closer to this goal with two key insights: a) sign alignment is enough for training a sparse initialized network and b) reparameterization can improve sign alignment.

## 8 Acknowledgements

We gratefully acknowledge the Gauss Centre for Supercomputing e.V. for funding this project by providing computing time on the GCS Supercomputer JUWELS at Jülich Supercomputing Centre (JSC) Jülich Supercomputing Centre (2021). We are also grateful for funding from the European Research Council (ERC) under the Horizon Europe Framework Programme (HORIZON) for proposal number 101116395 SPARSE-ML.

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

# A  Proof of Theorem 5.2

In this section we show the main theoretical result of the paper. We drop the dependence on the index for $w_1$ as we are only concerned with the one dimensional case here.

**Theorem A.1.** *Let $\tilde{f}$ be the teacher and $f$ be the student network such that $a$ and $w$ follow the gradient flow dynamics in Eq. (3) with a random balanced parameter initialization. Moreover, let $\beta_1 > \beta_2 > 0$. If $w_{in} > 0$, then $f$ can learn the correct target with probability $1 - \left(\frac{1}{2}\right)^n$. In the other case ($w_{in} \leq 0$) learning fails.*

Proof. The proof idea is to show that for a balanced initialization with $w_{in} > 0$ the flow for $t > 0$ always enters the open set

$$\Gamma_0 := \{(a, w) \in \mathbb{R}^2 : a < -w, w > 0\}.$$

Furthermore, we show that the flow stays in the open set $\Gamma_0$. The system is a Riemannian gradient flow which implies that the flow converges towards a stationary point in $\Gamma_0$. It remains to be shown that the stationary point at the origin is a saddle and the stable manifold of the origin is not in $\Gamma_0$. Thus, the remaining stationary points are the global optimizers.

First we show that for balanced initializations $|a_{in}| = w_{in} > 0$ enter the region $\Gamma_0$. In case $a_{in} = w_{in} > 0$, we have $(a_{in}, w_{in}) \in \Gamma_0$. In case $-a_{in} = w_{in} > 0$, we have that $(a_{in}, w_{in}) \in \bar{\Gamma}_0 \setminus \Gamma_0$ i.e. the boundary of $\Gamma_0$. Therefore, we need to show the gradient field at $(a_{in}, w_{in})$ points into $\Gamma_0$.

The balanced initialization implies that

$$\partial_a L(a_{in}, w_{in}) = -\partial_w L(a_{in}, w_{in}).$$

Moreover, since $\tilde{a} > 0$, $\partial_a L(a_{in}, w_{in}) > 0$. Using that $\beta_1 > \beta_2 > 0$ we have that the gradient field satisfies:

$$
\begin{aligned}
da_{in} &= -\sqrt{a_{in}^2 + \beta_1}\partial_a L(a_{in}, w_{in})dt \\
&= \sqrt{w_{in}^2 + \beta_1}\partial_w L(a_{in}, w_{in})dt \\
&< \sqrt{w_{in}^2 + \beta_2}\partial_w L(a_{in}, w_{in})dt = -dw_{in}
\end{aligned}
$$

Therefore there is a $t_0 > 0$ such that $a_{t_0} < -w_{t_0} < 0$. Thus there is a $t_0 > 0$ such that $(a_{t_0}, w_{t_0}) \in \Gamma_0$.

We have entered the set $\Gamma_0$, we have to show that we cannot leave the set $\Gamma_0$. This can be shown by computing the gradient field at the boundaries. The boundary can be split up into three cases:

- $B_1 := \{(a, w) \in \mathbb{R}^2 : -a = w > 0\}$
- $B_2 := \{(a, w) \in \mathbb{R}^2 : w = 0, a > 0\}$
- The origin $\{(0, 0)\}$

The first case of $B_1$ is covered by the balanced initialization. For the second case of $B_2$ we can compute the gradient field again. We now only need that $dw_t > 0$. We linearize $dw_t$:

$$dw_t = C\sqrt{\beta_2}a_t dt > 0,$$

where $C = \frac{1}{n}\sum_{i=1}^n \max\{0, z_i\}^2 > 0$ with probability $1 - \left(\frac{1}{2}\right)^n$. The last case is the saddle point at the origin which we show is not possible to be reached from the open set $\Gamma_0$. Thus for all $(a_{in}, w_{in}) \in \Gamma_0$ we have that for all $t \geq 0$, $(a_t, w_t) \in \Gamma_0$ or $\lim_{t\to\infty}(a_t, w_t) = (0, 0)$.

In the case that $w > 0$ the flow can be written as a dynamical system on a Riemannian manifold. This allows us to guarantee convergence to a stationary point. The flow is given by

$$
\begin{cases}
da_t = -C\sqrt{a_t^2 + \beta_1}\left(a_t w_t^2 - w_t\right)dt & a_0 = a_{\text{in}} \\
dw_t = -C\sqrt{w_t^2 + \beta_2}\left(a_t^2 w_t - a_t\right)dt & w_0 = w_{\text{in}},
\end{cases}
$$

where $C = \frac{1}{n}\sum_{i=1}^n \max\{0, z_i\}^2 > 0$ with probability $1 - \left(\frac{1}{2}\right)^n$. This dynamical system has stationary points at the origin and the set $aw = 1$. The dynamical system is a Riemannian gradient

flow system therefore the flow converges to a stationary point. The stationary point at the origin is a saddle point. Therefore, the only way of getting stuck at the origin is when we initialize on the associated stable manifold. We show that this not possible for the balanced initialization. We calculate the linearization of the stable manifold and use that the balanced initialization stays in $\Gamma_0$. The linearization at the origin $(0,0)$ is given by

$$\begin{cases} da_t = C\sqrt{\beta_1}w_t dt \\ dw_t = C\sqrt{\beta_2}a_t dt. \end{cases}$$

By a direct calculation of the eigen vectors the linearization of the stable manifold is given by the vector $\left(-\sqrt{\frac{|\beta_1|}{|\beta_2|}}, 1\right)$. It follows from $\beta_1 > \beta_2 > 0$ that the linearization of the stable manifold is outside of $\Gamma_0$. Suppose that from $\Gamma_0$ the stable manifold is reachable. Then there is a continuous differentiable curve $\gamma_t$ with initialization $\gamma_0 = (a_{in}, w_{in}) \in \Gamma_0$ such that $\lim_{t\to\infty} \gamma_t = (0,0)$. This is not possible as it violates the gradient field at the boundaries of $\Gamma_0$. Thus, the flow does not converge to the stationary point at the origin. This concludes the first part, since the only set of stationary points are the set of global optima.

The other two remaining cases fail as the boundary at $w = 0$ is not differentiable and the gradient flow stops there. $\square$

# B  *Sign-In* algorithm

We provide pseudocode for *Sign-In* for in Section 4 with rescaling as explained in Section 4.

---

**Algorithm 1** Sign-In

---

**Input:** objective $L$, scaling $\beta$, frequency $p$, epochs $T$, mask $S$, stop rescaling epoch $T_2$
Initialize $L(S \odot m_0 \odot w_0)$ such that $m_0 \odot w_0 = \theta_0$ and $m_0^2 - w_0^2 = \beta \mathbf{I}$.
**for** $i = 1$ **to** $T$ **do**
    **if** $i$ mod $p == 0$ and $i < T_2$ **then**
        Rescale $m_i \odot w_i = \theta_i$ such that $m_i^2 - w_i^2 = \beta \mathbf{I}$
    **end if**
    $m_{i+1}, w_{i+1} = \text{Optimizer}\left(L\left(S \odot m_i \odot w_i\right)\right)$
**end for**
**Return** Model parameters $\theta_T = S \odot m_T \odot w_T$

---

# C  Algorithm Initialization

We derive here an exact analytic formula for the initialization of the reparameterization. Since the reparameterization is pointwise we can restrict ourself to the one dimensional case. For this, we need to solve the system of equations for arbitrary $x \in \mathbb{R}$ and $\beta > 0$:

$$\begin{cases} m_0 w_0 = x \\ m_0^2 - w_0^2 = \beta \end{cases}$$

We find that a solution is given by:

$$m_0 = u \text{ and } w_0 = \frac{x}{u}$$

where $u = \sqrt{\alpha + \sqrt{x^2 + \alpha^2}}$, with $\alpha = \frac{\beta}{2}$. Note that the solution can also be written as

$$m_0 = \frac{v + \frac{\alpha}{v}}{\sqrt{2}} \text{ and } w_0 = \frac{v - \frac{\alpha}{v}}{\sqrt{2}}$$

where $v = \sqrt{x + \sqrt{x^2 + \alpha^2}}$ with $\alpha = \frac{\beta}{2}$.

# D Multiple Neurons Visualization

We also explore the multi neuron case. The neural network $f_2 : \mathbb{R}^k \times \mathbb{R}^{kd} \times \mathbb{R}^d \to \mathbb{R}$ is given by $f_2(\mathbf{a}, \mathbf{W}, \mathbf{z}) := \sum_{i=1}^k a_i \sigma \left( \mathbf{w}_i^T z \right)$. The data is generated from a similar structured teacher network $\tilde{f}_2$ with $k = 3$ neurons and input dimension $d = 2$. Furthermore, $\tilde{\mathbf{a}} \sim Uni(\{-1, 1\})$ i.e. i.i.d. Rademacher random variables and $\tilde{\mathbf{w}}_{i,j} \sim N(0, 1)$ for all $i, j \in [k, d]$ and normalized over the input dimension. We initialize the dense network with $k = 20$ and the so-called COB initialization based on the rich regime in (Chizat et al., 2020). All weights are initialized with $\mathcal{N}(0, 1/d)$ and we ensure that $a_i = -a_{i+10}$ for $i \in [10]$. In the sparse cases we initialize $k = 3$ neurons. In the good signs case we initialize such that the signs of the weights align with the teacher network. In contrast for the bad signs we choose $a_i > 0$ for all $i \in [3]$. The learning rate used is $\eta = 1$ for the sparse setup and $\eta = 2.15$ for the dense setup. Moreover, the scaling for the reparameterization is $\beta = 2$. This results in Figure 5, where we represent each neuron as $|a_i|\mathbf{w}_i$. The teacher neurons are vectors touching the unit circle and student neurons are denoted with markers. We illustrate both the benefit of correct sign alignment (Fig. 5(b)) and the reparameterization in case of bad alignment (Fig. 5(d)). This is contrasted with dense training (Fig. 5(a)) and bad alignment (Fig. 5(c)).

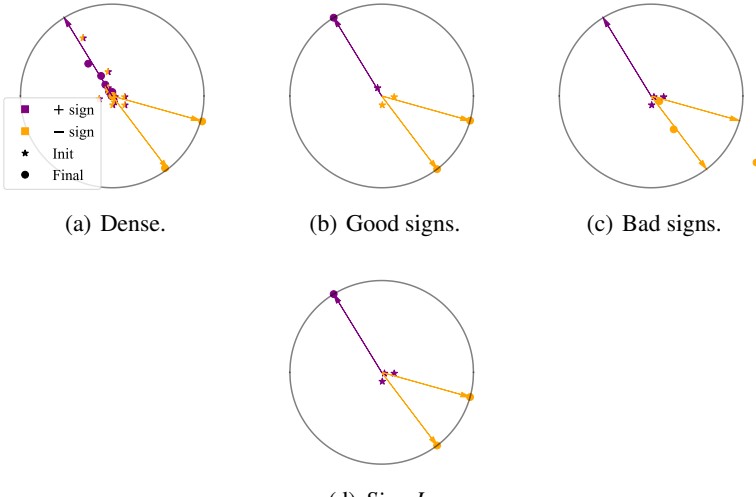

(a) Dense.      (b) Good signs.      (c) Bad signs.

(d) *Sign-In.*

Figure 5: Representation of a two layer student-teacher neural network inspired by Chizat et al. (2020). Signs are crucial in learning sparse representations. A dense neural network can easily learn the representation in Fig. 5(a). Sparse networks can learn when signs are correct Fig. 5(b) and fail with bad signs Fig. 5(c). The reparameterization enables learning the representation with bad signs in Fig. 5(d).

# E  Training Details

In this section we provide the details of the additional hyperparameters used for experiments. Table 4 we provide the standard hyperparameters such as learning rate and weight decay. We note that the loss used for both CIFAR10 and CIFAR100 is the cross-entropy loss and for ImageNet the label smoothing loss with confidence 0.9. For the $m \odot w$ parametrization we also apply Frobenius decay i.e. $(m \odot w)^2$ with the same strength as weight decay for CIFAR10 and CIFAR100. We provide these extra details in Table 5. The code used is based on TurboPrune as in (Nelaturu et al.).

Table 4: Training Details for all experiments presented in the paper.

| Dataset | Model | LR | WD | Epochs | Batch Size | Optim | Schedule |
|---------|-------|-----|------|--------|------------|-------|----------|
| CIFAR10 | ResNet20 | 0.2 | $1e-4$ | 150 | 512 | SGD | Triangular |
| CIFAR100 | ResNet18 | 0.2 | $1e-4$ | 150 | 512 | SGD | Triangular |
| ImageNet | ResNet50 | 0.25 | $5e-5$ | 100 | 1024 | SGD | Triangular |
|          | DeIT Small | 0.001 | $1e-1$ | 300 | 1024 | AdamW | Triangular |

Table 5: Training Details for the additional *Sign-In* parameters. The weight decay in this table overides the weight decay in Table 4.

| Dataset | Model | $\beta$ | $T_2$ | Frobenius Decay | Weight decay |
|---------|-------|---------|-------|-----------------|--------------|
| CIFAR10 | ResNet20 | 1 | 75 | $1e-4$ | 0 |
| CIFAR100 | ResNet18 | 1 | 75 | $1e-4$ | 0 |
| ImageNet | ResNet50 | 1 | 75 | 0 | $5e-5$ |
|          | DeIT Small | 1 | 225 | 0 | $1e-1$ |

# F  Experiments on two-layer networks with *Sign-In*

We empirically verify the ability to recover correct signs for dense-to-sparse methods and *Sign-In* on the two-layer network with a student-teacher setup described in Section 5. Further, we also empirically evaluate the effect of combining dense-to-sparse methods with *Sign-In* on the two-layer network. We repeat 100 runs for each case, with data generated from a teacher $\tilde{a} = 1, \tilde{w} = [1, 0, .., 0]$, such that the labels are $y = \tilde{f}_0(\tilde{a}, \tilde{w}, z)$. For the multidimensional case we use $d = 5$. Each setting is trained with Gradient Descent for $2e - 5$ iterations with a learning rate $= 0.01$. Results reported in Table 6 are the success fraction over 100 runs. We empirically observe that it is possible to successfully recover all cases when combining overparameterized training with *Sign-In*. Even the most difficult case that is unreachable for both on their own i.e. $a < 0$ and $w < 0$. Now it is possible due to *Sign-In* being able to flip $a$, whereas overparameterized training can flip $w$ after that. This is also observed with improved performance of AC/DC with *Sign-In* (see Table 11).

Table 6: Experiments on two-layer networks in the student-teacher setup presented in Theorem 5.2.

| Method | Initial Signs | | | |
|---|---|---|---|---|
| | $a > 0, w > 0$ | $a < 0, w > 0$ | $a > 0, w < 0$ | $a < 0, w < 0$ |
| Sparse | 1 | 0 | 0 | 0 |
| Overparam | 1 | 0 | 1 | 0 |
| *Sign-In* | 1 | 1 | 0 | 0 |
| Overparam + *Sign-In* | 1 | 1 | 1 | 1 |

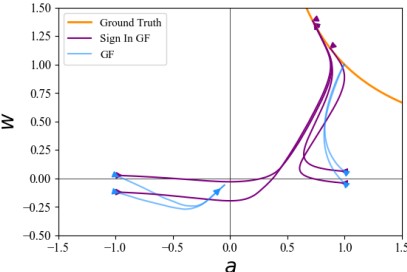

Figure 6: Student-teacher setup: training one neuron with multidimensional inputs $d = 5$. Combining the overparameterized case with *Sign-In* can recover all cases .

**Failure case for Overparam + *Sign-In***   We find that Overparam + *Sign-In* does not always succeed in recovering all the cases, and requires sufficient overparameterization to correctly flip $a < 0$.

We choose the teacher network $\tilde{f}_0(\tilde{a}, \tilde{w}, z)$ with $\tilde{a} = 1$ and $\tilde{\mathbf{w}}_{\mathbf{ij}} \sim N(0, 1)$, and train the student with $d = 2$ for the case with $a < 0$. We find that Overparam + *Sign-In* is not able resolve this case every time and requires additional overparameterization to improve success rates.

# G  *Sign-In* flips signs in later layers

Here we validate that the sign flips enabled by *Sign-In* are different from those enabled with dense-to-sparse training. For the case $s = 0.9$ reported in Table 11 for AC/DC and AC/DC + *Sign-In*, we plot the layerwise sign flips for three stages, initialization to warmup (10-th epoch), warmup to final and initialization to final. Additionally, we also compare the layerwise sign flips of sparse training a randomly reinitialized AC/DC mask. While AC/DC identifies the correct sign flips upto warmup, *Sign-In* enables more sign flips during subsequent training for later layers in the network (see orange line in Figure 7(b)).

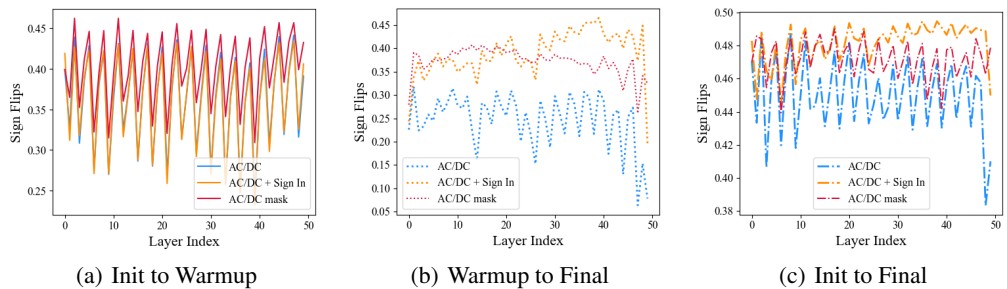

(a) Init to Warmup          (b) Warmup to Final          (c) Init to Final

Figure 7: Layerwise sign flips for *Sign-In* for AC/DC, AC/DC + *Sign-In* and sparse training with an AC/DC mask on a ResNet50 with $90\%$ trained on ImageNet.

# H  *Sign-In* with other PaI methods

We report sparse training performance with *Sign-In* across other PaI methods including SNIP (Lee et al., 2019) and Synflow (Tanaka et al., 2020) in Table 16. We use the same setup as described in Appendix E.

We observe in Table 16 that *Sign-In* improves all PaI methods. Moreover, in most cases the random mask performs best over all.

Table 7: *Sign-In* with PaI methods on CIFAR10 and CIFAR100. ($*$ denotes a single run, as the other seeds crashed).

| Dataset | Method | $s = 0.8$ | $s = 0.9$ | $s = 0.95$ |
|---|---|---|---|---|
| CIFAR10 $_{+\,\text{ResNet20}}$ | Random | $88.41(\pm0.23)$ | $86.41(\pm0.1)$ | $83.62(\pm0.28)$ |
| | Random + *Sign-In* | $89.37(\pm0.14)$ | $\mathbf{87.83(\pm0.11)}$ | $\mathbf{84.74(\pm0.06)}$ |
| | SNIP | $88.12(\pm0.5)$ | $85.82(\pm0.2)$ | $81.98(\pm0.32)$ |
| | SNIP + *Sign-In* | $\mathbf{89.88(\pm0.09)}$ | $87.27(\pm0.13)$ | $83.37(\pm0.06)$ |
| | Synflow | $88.15(\pm0.17)$ | $85.32(\pm0.24)$ | $73.22^*$ |
| | Synflow + *Sign-In* | $89.26(\pm0.05)$ | $86.47(\pm0.06)$ | $80.13^*$ |
| | NPB | $87.30(\pm0.39)$ | $85.98(\pm0.32)$ | $82.06(\pm0.9)$ |
| | NPB + *Sign-In* | $88.87(\pm0.02)$ | $86.88(\pm0.51)$ | $83.37(\pm0.65)$ |
| CIFAR100 $_{+\,\text{ResNet18}}$ | Random | $74.01(\pm0.1)$ | $73.07(\pm0.19)$ | $71.41(\pm0.18)$ |
| | Random + *Sign-In* | $75.32(\pm0.25)$ | $73.94(\pm0.13)$ | $72.51(\pm0.14)$ |
| | SNIP | $74.08(\pm0.28)$ | $72.73(\pm0.32)$ | $71.21(\pm0.15)$ |
| | SNIP + *Sign-In* | $75.16(\pm0.15)$ | $73.64(\pm0.32)$ | $72.03(\pm0.50)$ |
| | Synflow | $73.41(\pm0.28)$ | $71.37(\pm0.42)$ | $68.56(\pm0.17)$ |
| | Synflow + *Sign-In* | $75.16(\pm0.20)$ | $73.64(\pm0.33)$ | $72.03(\pm0.70)$ |
| | NPB | $74.92(\pm0.63)$ | $72.79(\pm0.56)$ | $71.49(\pm0.24)$ |
| | NPB + *Sign-In* | $\mathbf{75.59(\pm0.15)}$ | $\mathbf{74.72(\pm0.18)}$ | $\mathbf{73.28(\pm0.4)}$ |

**Different masks**  We perform sparse training with a random reinitialization of masks identified by dense-to-sparse and compare them with PaI methods. Once reinitialized, dense-to-sparse masks loose alignment with parameters and are at par with PaI methods (Jain et al., 2024; Frankle et al., 2021). These experts reveal two key insights:

- Layerwise sparsity ratios determine trainiability of sparse networks in the absence of sign alignment.
- *Sign-In* benefits from low sparsity in initial layers.

First, we observe that in case of masks identified by STR and PaI with Snip and Synflow achieve poor performance for both standard training and *Sign-In*. The layerwise sparsity ratios of these masks reveal sparse initial layers ($> 90\%$) or highly sparse ($> 99\%$) later layers, which potentially affect the overall trainability of the network. In contrast, the RiGL mask, which is a random mask with balanced layerwise sparsity ratio (Gadhikar et al., 2023), performs the best over all other masks with *Sign-In*. Here initial layers are dense while later layers always have sparsity $< 99\%$.

Second, we observe that when initial layers have low sparsity, *Sign-In* can outperform standard training (RiGL, AC/DC, Snip) but struggles if this is not the case (STR, Synflow). A potential cause would be the inability of initial layers to flip $w$ (see Theorem 5.2).

Table 8: *Sign-In* with different masks for a ResNet50 trained on ImageNet with $90\%$ sparsity and random initialization. ($*$ denotes a single run, as the runs for other seeds crashed.

| Mask | Init | |
|---|---|---|
| | Random | Random + *Sign-In* |
| AC/DC | $70.66_{\pm 0.12}$ | $70.96_{\pm 0.09}$ |
| RiGL | $72.02_{\pm 0.23}$ | $\mathbf{72.48}_{\pm \mathbf{0.19}}$ |
| STR | $68.36_{\pm 0.17}$ | $67.81_{\pm 0.34}$ |
| Snip | $52.9^*$ | $54.27^*$ |
| Synflow | $60.66_{\pm 0.2}$ | $60.59_{\pm 0.07}$ |
| Random | $71.56_{\pm 0.03}$ | $72.19_{\pm 0.18}$ |

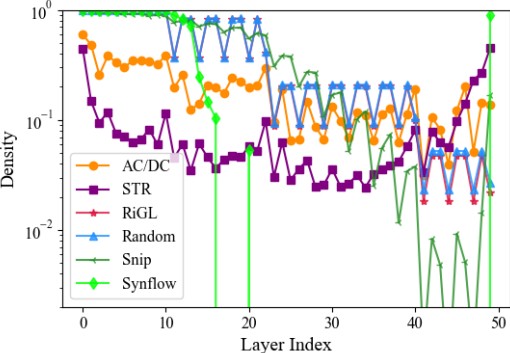

Figure 8: Layerwise sparsity ratios at $90\%$ sparsity for a ResNet50.

## I  Sign Importance

In order to identify the amount of correct signs required for training with the sparse mask to match the respective dense-to-sparse baseline (via sign-alignment), we perturb an increasing fraction of signs as reported in Table 9. Perturbing more than $10\%$ of the signs starts degrading performance rapidly.

Table 9: Ablation study on the importance of weight signs.

| Method | $s = 0.9$ |
|---|---|
| AC/DC baseline (from Table 1) | 74.68 |
| AC/DC Mask + $100\%$ correct signs (from Table 1) | 74.88 |
| AC/DC Mask + $95\%$ correct signs | 74.44 |
| AC/DC Mask + $90\%$ correct signs | 73.54 |
| AC/DC Mask + $80\%$ correct signs | 71.81 |
| AC/DC Mask + random signs (from Table 1) | 70.88 |

## J  Vision transformer with *Sign-In*

*Sign-In* is also able to improve sparse training of a randomly masked vision transformer. A DeiT Small architecture (trained for 300 epochs from scratch) gains more than $1\%$ in performance with *Sign-In*, as reported in Table 10.

## K  *Sign-In* improves other sparse training methods

Only a combination of *Sign-In* and overparameterization is able to solve all sign flip challenges in our theoretical setting, as also confirmed by numerical experiments, which are provided in Table 6

Table 10: **DeiT Small** on ImageNet, trained from scratch.

| Method | $s = 0.5$ |
|---|---|
| Random | $69.31(\pm 0.7)$ |
| Random + *Sign-In* (ours) | $\mathbf{70.62(\pm 0.4)}$ |

in Appendix F. Therefore, we expect the combination of dense-to-sparse methods with *Sign-In* to achieve further gains in performance. Table 11 confirms that AC/DC improves with *Sign-In*. However, the improvement is less significant compared to PaI.

Table 11: *Sign-In* **improves AC/DC** for a ResNet50 trained on ImageNet.

| Method | $s = 0.8$ | $s = 0.9$ | $s = 0.95$ |
|---|---|---|---|
| AC/DC | $75.83(\pm 0.02)$ | $74.75(\pm 0.02)$ | $72.59(\pm 0.11)$ |
| AC/DC + *Sign-In* | $\mathbf{75.9(\pm 0.14)}$ | $74.74(\pm 0.12)$ | $\mathbf{72.88(\pm 0.13)}$ |

*Sign-In* also improves dynamic sparse training with RiGL and MEST by enabling better sign flips.

Table 12: *Sign-In* **improves RiGL** for a ResNet50 trained on ImageNet.

| Method | $s = 0.8$ | $s = 0.9$ | $s = 0.95$ |
|---|---|---|---|
| RiGL | $75.02(\pm 0.1)$ | $73.7(\pm 0.2)$ | $71.89(\pm 0.07)$ |
| RiGL + *Sign-In* | $75.02(\pm 0.1)$ | $\mathbf{74.27(\pm 0.08)}$ | $\mathbf{73.07(\pm 0.17)}$ |

Table 13: *Sign-In* **improves MEST** for a ResNet50 trained on ImageNet.

| Method | $s = 0.8$ | $s = 0.9$ | $s = 0.95$ |
|---|---|---|---|
| MEST | 74.67 | 73.1 | 70.84 |
| MEST + *Sign-In* | **74.74** | **73.32** | **71.41** |

## L   Rescaling $m \odot w$ in *Sign-In*

While our proposed pointwise reparameterization with $m \odot w$ allows better sign flips for sparse training, we find that the rescaling step outlined in Appendix B is essential to maximize the benefits of the reparameterization as show with improved performance of *Sign-In* over $m \odot w$ in Table 14.

Table 14: Effect of Rescaling on *Sign-In*.

| Dataset | Method | $s = 0.8$ | $s = 0.9$ | $s = 0.95$ |
|---------|--------|-----------|-----------|------------|
| CIFAR10 $_{+ \text{ResNet20}}$ | Random | $88.41(\pm 0.23)$ | $86.41(\pm 0.1)$ | $83.62(\pm 0.28)$ |
| | Random + *Sign-In* | $\mathbf{89.37}(\pm \mathbf{0.14})$ | $\mathbf{87.83}(\pm \mathbf{0.11})$ | $\mathbf{84.74}(\pm \mathbf{0.06})$ |
| | Random + $m \odot w$ | $89.11(\pm 0.08)$ | $87.13(\pm 0.23)$ | $83.88(\pm 0.02)$ |
| CIFAR100 $_{+ \text{ResNet18}}$ | Random | $74.01(\pm 0.1)$ | $73.07(\pm 0.19)$ | $71.41(\pm 0.18)$ |
| | Random + *Sign-In* | $\mathbf{75.32}(\pm \mathbf{0.25})$ | $\mathbf{73.94}(\pm \mathbf{0.13})$ | $\mathbf{72.51}(\pm \mathbf{0.14})$ |
| | Random + $m \odot w$ | $74.62(\pm 0.24)$ | $73.18(\pm 0.01)$ | $72.08(\pm 0.25)$ |
| ImageNet $_{+ \text{ResNet50}}$ | Random | $73.87(\pm 0.1)$ | $71.55(\pm 0.04)$ | $68.69(\pm 0.03)$ |
| | Random + *Sign-In* | $\mathbf{74.12}(\pm \mathbf{0.09})$ | $\mathbf{72.19}(\pm \mathbf{0.18})$ | $\mathbf{69.38}(\pm \mathbf{0.1})$ |
| | Random + $m \odot w$ | $73.96(\pm 0.04)$ | $71.83(\pm 0.17)$ | $69.15(\pm 0.09)$ |

# M  Code for *Sign-In* in PyTorch

We provide an example of integrating *Sign-In* in any training setup in PyTorch by simply replacing the Conv (or Linear) layer with 1.

```python
import torch
import torch.nn as nn

class ConvMaskMW(nn.Conv2d):
    """
    Conv2d layer with a parametrization of m*w for the weights.
    Additionally a mask is applied during the forward pass to the weights of the layer.

    Args:
        **kwargs: Keyword arguments for nn.Conv2d.
    """
    def __init__(self, **kwargs):
        super().__init__(**kwargs)
        self.register_buffer("mask", torch.ones_like(self.weight))

        self.alpha = alpha = 0.5
        u = torch.sqrt(self.weight.to(self.weight.device) + torch.sqrt(self.weight.to(
            self.weight.device)*self.weight.to(self.weight.device) + alpha*alpha))

        self.m =  nn.Parameter((u + alpha/u)/np.sqrt(2), requires_grad=True)
        self.weight.data = (u - alpha/u)/np.sqrt(2)

    def merge(self):
        # This method combines m and w to give an effective weight, for inference or
            during pruning.
        self.weight.data = self.m.to(self.weight.device) * self.weight

    def rescale_mw(self):
        x = self.m.to(self.weight.device) * self.weight.to(self.weight.device)
        u = torch.sqrt(self.alpha + torch.sqrt(x*x + self.alpha*self.alpha))
        self.m.data = u
        self.weight.data = x/u

    def forward(self, x):
        """Forward pass with masked weights."""

        sparseWeight = self.mask.to(self.weight.device) * self.m.to(self.weight.device) *
            self.weight.to(self.weight.device)
        return F.conv2d(
            x,
            sparseWeight,
            self.bias,
            self.stride,
            self.padding,
            self.dilation,
            self.groups,
        )
```

Listing 1: Conv2d layer with *Sign-In* parametrization of $m \odot w$.

# N  FLOPS required for *Sign-In*

We perform a FLOP comparison for Sign-In vs. other PaI methods. The FLOP overhead due to the reparameterization in Sign-In is negligible and only required during training, as $m \odot w$ can be merged after training, during inference.

For example, the FLOPs for a convolutional layer are:

$$(2 \times H_{\text{out}} \times W_{\text{out}} \times C_{\text{out}} \times K \times K \times C_{\text{in}}) + (C_{\text{out}} \times C_{\text{in}} \times K \times K)$$

An additional term $(C_{\text{out}} \times C_{\text{in}} \times K \times K)$ is introduced by the elementwise product $m \odot w$, but it is negligible with respect to the other term.

We report the FLOPs per forward pass for a ResNet50.

Table 15: **FLOP Comparison** of Sign-In vs. other PaI methods on ResNet50. Sign-In introduces minimal overhead while improving accuracy.

| Method | Training FLOPs (G) | Inference FLOPs (G) | Acc |
|---|---|---|---|
| Dense | 8.2 | 8.2 | 76.89 |
| Sparse ($s = 0.8$) | 4.99 | 4.99 | 73.87 |
| Sign-In ($s = 0.8$) | 5 | 4.99 | 74.12 |
| Sparse ($s = 0.9$) | 3.89 | 3.89 | 71.56 |
| Sign-In ($s = 0.9$) | 3.9 | 3.89 | 72.19 |

# O  High sparsity regime ablation

In this section we provide results for the high sparsity regime.

Table 16: *Sign-In* with PaI methods on CIFAR10 and CIFAR100 in the high sparsity regime.

| Dataset | Method | $s = 0.98$ | $s = 0.99$ |
|---|---|---|---|
| CIFAR10  + ResNet20 | Random | $88.41(\pm0.23)$ | $86.41(\pm0.1)$ |
| | Random + *Sign-In* | $89.37(\pm0.14)$ | $\mathbf{87.83(\pm0.11)}$ |
| | SNIP | $88.12(\pm0.5)$ | $85.82(\pm0.2)$ |
| | SNIP + *Sign-In* | $\mathbf{89.88(\pm0.09)}$ | $87.27(\pm0.13)$ |
| | Synflow | $88.15(\pm0.17)$ | $85.32(\pm0.24)$ |
| | Synflow + *Sign-In* | $89.26(\pm0.05)$ | $86.47(\pm0.06)$ |
| CIFAR100 + ResNet18 | Random | $74.01(\pm0.1)$ | $73.07(\pm0.19)$ |
| | Random + *Sign-In* | $\mathbf{75.32(\pm0.25)}$ | $\mathbf{73.94(\pm0.13)}$ |
| | SNIP | $74.08(\pm0.28)$ | $72.73(\pm0.32)$ |
| | SNIP + *Sign-In* | $75.16(\pm0.15)$ | $73.64(\pm0.32)$ |
| | Synflow | $73.41(\pm0.28)$ | $71.37(\pm0.42)$ |
| | Synflow + *Sign-In* | $75.16(\pm0.20)$ | $73.64(\pm0.33)$ |

