# OpenReview forum: "Sign-In to the Lottery: Reparameterizing Sparse Training"
_NeurIPS.cc/2025/Conference — NeurIPS 2025 poster_

### Official Review · Reviewer_DCG2 · 2025-06-04

**Clarity:** 3
**Significance:** 3
**Originality:** 3
**Rating:** 4
**Confidence:** 5

**Summary:**

This paper explores the role of sign-flipping dynamics in sparse training, proposing a reparameterization approach to enhance early, stable sign alignment. The authors argue that such alignment is a crucial factor distinguishing dense-to-sparse methods from sparse-from-scratch approaches. Their empirical findings demonstrate substantial performance gains in classification tasks across CIFAR-10, CIFAR-100, and ImageNet, while a theoretical analysis of a toy model further reinforces the connection between sign flips and convergence in overparameterized networks.

**Questions:**

In Table 1, please confirm that the reported numbers are indeed the final Top-1 classification accuracies for ImageNet with 90% sparsity.

Could you elaborate on the behavior of Equation (1) (See weaknesses)?

Do you have experiments that show the behavior for different values of $\beta$?

​While the paper shows that early sign alignment correlates with better performance, what happens when you directly manipulated sign flips to see how it affects accuracy?

**Ethical Concerns:**

["NO or VERY MINOR ethics concerns only"]

**Final Justification:**

I think it's a really nice paper. Based on the rebuttal I'm happy to change to accept.

**Limitations:**

Yes

**Paper Formatting Concerns:**

no concerns

**Quality:**

3

**Strengths And Weaknesses:**

$\textbf{STRENTHS}$

-Novel Insight: The paper offers an interesting and novel perspective on why dense-to-sparse methods outperform sparse-from-scratch, highlighting the role of early sign alignment in sparse networks.

-Empirical Validation: The extensive experiments across multiple datasets convincingly support the claim that early sign flipping is critical for performance in sparse settings.

-Theoretical Support: The use of a toy single-neuron model in Theorem 5.1 provides a clear and intuitive illustration of the sign-flipping dynamics and their potential benefits.

-Clarity in Main Sections: The introduction and main experimental results are generally well written and easy to follow.

$\textbf{WEAKNESSES}$

-Clarity of Key Results: Figure 1 does not clearly state that it reports sign flip dynamics, not accuracy values, which could confuse readers. Additionally, Table 1 does not explicitly label its entries as final Top-1 classification accuracies on ImageNet (90% sparsity), making it harder to interpret.

-Classification of Methods: The distinction between dense-to-sparse and sparse-from-scratch methods in Figure 1 (the first three bars vs. the rest) is not explicitly named, limiting interpretability.

-Acronym Usage: The acronym “PaI” is introduced in the title and abstract without being defined, which could be confusing for readers unfamiliar with the term.

-Sign Flipping Mechanism (Equation 1): There seems to be a potential issue when the parameter $\phi_t$  is large and negative, and the gradient is positive. In this case, the reparameterized update direction appears to move  $\phi_t$
  further negative, which seems counter to the intended sign-flipping mechanism. Clarification on how this behavior is mitigated in practice would strengthen the paper’s claims.

-Single Neuron Clarification: In Theorem 5.1 and its discussion, it would be helpful to explicitly state that both the student and teacher networks are single-neuron (1–1–1) networks, to avoid any ambiguity and help readers understand the simplicity of the toy model used.

-I feel like there would have been better experiments than the one in Fig 1 (see questions section).

-The paper has very little theory.

---

> ### Author Rebuttal · Authors · 2025-07-30
>
> We would like to express our gratitude for your time and efforts in providing valuable comments on our manuscript. We appreciate your positive comments regarding the clarity of our work, theoretical support and extensive experiments. Please find a detailed point-by-point response to your comments below. In case of any open questions, we would be happy to discuss them.
>
> **Rebuttal of Weaknesses**
>
> 1. In the revised manuscript, we would be happy to extend the table and figure description as suggested to highlight sign flipping dynamics and that the reported results are for Top-1% accuracy on ImageNet.
>
> 2. To make the distinction between DST and PaI in Figure 1 clearer, we will add two headers saying DST and PaI. Moreover, we will make the current vertical line that indicates the distinction bolder.
>
> 3. We agree with the reviewer and will replace “training sparse neural networks from scratch (PaI)” with the full name Pruning at Initialization (PaI) in the abstract and define it early on.
>
> 4. If the gradient is positive ($\nabla_{\theta}f > 0$) and the parameter theta is negative ($\theta < 0$), this means that the sign is perceived to be already correct (assuming that the gradient is pointing in the correct direction). This is also true for standard gradient descent. Note that Sign-In controls the relative magnitude but not the direction itself. In practice, we would work with stochastic gradient estimates. Noisy gradients could push the parameter away from zero wrongly then, as in case of standard gradient descent. Sign-In would simply not add merit in this case.
> The main benefit of Sign-In is realized in the case when the gradient and parameter are both pointing in the same direction (eg: $\nabla_{\theta}f > 0, \theta > 0$), which would imply that the gradient is pushing the parameter towards 0 (because $\theta \leftarrow \theta - \eta \nabla_{\theta}f$). Sign-In speeds up this movement towards 0 and promotes thus a sign flip.
>
> 5. We will explicitly state that both the teacher and student are single neuron neural networks in a revision of our manuscript.
>
> 6. To strengthen the our conclusions from Figure 1, we provide additional experiments by directly manipulating the sign as suggested by the reviewer. We also report results where Sign-In improves state-of-the-art pruning methods like dynamic sparse training with RiGL. See tables below.
>
> 7. Our theory combines insights from two lines of work, namely, Riemannian gradient flows induced by overparameterization and dynamical systems theory. The combination of both is a novel idea and requires non-trivial proof steps. Moreover, the theory captures and provides insights into a novel practical algorithm, Sign-In, which translates to large scale experiments and presents a step forward towards solving the PaI problem. For that reason, we see potential that our conceptual insights could inspire further theoretical investigations and new method design.
>
> **Answers to Questions:**
>
> 1. Yes these are indeed the top 1% accuracy values. We will make sure that this is clear from the table description in the revised manuscript.
>
> 2. Please see Point 4 in weaknesses (above).
>
> 3. Different $\beta$ would lead to different effective learning rates (see Eq.(1) for example). Setting $\beta =1$ allows us to keep the effective learning rate close to the original learning rate of standard optimizers. To illustrate this point, we have conducted an experiment for different $\beta$ on CIFAR100 with a Resnet18 for random pruning with $90$% sparsity:
>
> | $\mathbf{\beta}$ | s=0.9 |
> |------------|---------------|
> | 0.2    	| 73.26 ± 0.17  |
> | 0.5    	| 73.95 ± 0.04 |
> | 1      	| 73.94 ± 0.13  |
> | 4      	| 73.99 ± 0.38  |
> | 20     	| 73.03 ± 0.29  |
>
> We observe that higher $\beta$ could lead to slightly better performance, most likely due to more opportunities for correct sign flips. Yet, it also induces a higher variance due to the higher effective learning rate. Across different experiments, $\beta=1$ required less tuning of other hyperparameters. Hence, we choose this value for $\beta$ in all our experiments.
>
> 4. In Table 1 in the main text, we have studied the effect of manipulating signs at initialization. In principle, it might also be possible to provide the right signs later during training, as parameters need to move less given the right signs. However, this is an orthogonal question to solving PaI.
>
> In addition, we report results for manipulating the signs as suggested by the reviewer, extending the results from Table 1 for ACDC. We find the reducing the number of correct signs progressively worsens performance.
>
> | Method                     	| s=0.9   |
> |--------------------------------|---------|
> | ACDC baseline (from Table 1) 	| 74.68	|
> | ACDC Mask + 100% correct signs (from Table 1) | 74.88 |
> | ACDC Mask + 95% correct signs  |  74.44 |
> | ACDC Mask + 90% correct signs  | 73.54    |
> | ACDC Mask + 80% correct signs  | 71.81 |
> | ACDC Mask + random sign (from Table 1)  | 70.88 |
>
> To strengthen our point that Sign-In improves contemporary sparse training methods, we provide experiments with RiGL + Sign-In. This highlights that learning with a dynamic mask as in RiGL, Sign-In also helps with the sign flipping mechanism, which can improve performance at high sparsity, especially.
>
> | Method     	| s = 0.8        	| s = 0.9           | s = 0.95          |
> |----------------|---------------|-------------------|------------------|
> | RiGL       	| 75.02 ± 0.1   | 73.70 ± 0.2   	| 71.89 ± 0.07 	|
> | RiGL + Sign-In | 75.02 ± 0.1   | 74.27 ± 0.085 	| 73.07 ± 0.17 	|
>
>
> We sincerely thank you for your thoughtful feedback and constructive suggestions, which have improved our paper. We are looking forward to the discussion period and would be happy to address further questions and provide additional clarifications.

---

> ### Author Response · Authors · 2025-08-05
> **Response to Reviewer Comments**
>
> We sincerely thank you for your thoughtful feedback and for engaging in this discussion. We are happy that we were able to address most of your comments and questions.
>
> To answer your question, our analysis relies on the standard assumption that the gradient points in the correct direction. This is a necessary precondition for any first-order optimizer—with or without Sign-In—to function correctly.
>
> Therefore, in the specific scenario you described (a large negative parameter with a positive gradient), the gradient correctly indicates that the parameter should become more negative. In this case, a sign flip is undesirable, and our method is designed not to force one.
>
> **Toy Example**: Let us make this more concrete with our toy example from Figure 3a, given a single neuron network with a scalar input and a ReLU activation, $f(x; a, w) = a \sigma (wx)$. This network, $f(x; a, w)$, is trained with gradient descent and LR=$0.005$. In our analysis, we have considered all four possible initializations for $a, w$ parameterized with Sign-In, with a balanced initialization ($a^2 = w^2$). We construct a case which generates a positive gradient for a parameter which is negative by assuming a ground truth $y = -10 \sigma (x)$ and discuss the four possible cases.
> Note that the cases when $w < 0$ will fail due to the ReLU activation which is in line with Figure 2 (Sparse + Sign-In case). Experiments for the remaining two other cases are presented by the next table.
>
> | Experiment | Initial $a_0$ | Initial $w_0$ | Gradient $\frac{\partial L}{\partial a}$ | Gradient $\frac{\partial L}{\partial w}$ | Final $a_{\text{learnt}}$ | Final $w_{\text{learnt}}$ | Final Product $a_{\text{learnt}}w_{\text{learnt}}$ |
> |:----------:|:-------------:|:-------------:|:--------------------------------------:|:--------------------------------------:|:---------:|:---------:|:--------------------:|
> | 1          | $2.0$         | $2.0$         | $>0$                                   | $<0$                                   | $-3.1954$ | $3.1296$  | $-10.0$              |
> | 2          | $-2.0$        | $2.0$         | $>0$                                   | $<0$                                   | $-3.1737$ | $3.1509$  | $-10.0$              |
>
> * In Experiment 1 (in the table above), the sign of $a$ is flipped by Sign-In to reach the ground truth. This would not happen without Sign-In.
> * Experiment 2 replicates the scenario described by you. In this case, $a$ does not need to switch its sign to reach the ground truth. With and without Sign-In, $a$ correctly converges to the ground truth and becomes more negative.
>
> In both cases, learning with Sign-In finds the ground truth, as verified by the final product $a_{\text{learnt}}w_{\text{learnt}} = -10$.
>
> In addition, note that Sign-In only modifies the magnitude of the gradient step by multiplying with $\sqrt{\theta^2 + \beta}$ to facilitate a sign flip but does not change the sign of the gradient of the original parameter to force a sign flip.
>
> **Regarding our comment on SGD**: The gradient updates used in practice are based on mini-batches, making them noisy approximations of the true, full-dataset gradient. As a result, the direction of these gradient estimates can sometimes be suboptimal. This is a universal property of SGD that affects all first-order methods, including our own, and is not a new limitation introduced by Sign-In.

---

> > ### Comment · Reviewer_DCG2 · 2025-08-05
> >
> > Thank you for your explanation, I found it very useful. No need to comment on this, but I think in early training when theta is large, noisy gradients might be amplified, but that's probably fine.
> >
> > I don't have any further questions.

---

> > > ### Author Response · Authors · 2025-08-05
> > > **Response to Reviewer**
> > >
> > > Thank you for the positive feedback; we are glad our explanation was useful.
> > >
> > > **Regarding noise gradients for large theta**: This effect is controlled by regularization with weight decay (also in the case of Sign-In). By penalizing large parameter values during training, regularization helps keep theta within a reasonable range, which in turn mitigates the potential amplification of noisy gradients.
> > >
> > > Thank you again for the insightful discussion.

---

### Official Review · Reviewer_G1i7 · 2025-07-01

**Clarity:** 2
**Significance:** 2
**Originality:** 2
**Rating:** 5
**Confidence:** 3

**Summary:**

The paper demonstrates the importance of learning signs in training a pruned model through theoretical analysis and experiments. It also proposes a method that promotes sign learning via reparameterization. Experiments show that this method improves performance across datasets and confirms that it encourages sign flips during training.

**Questions:**

As the authors mentioned, the proposed method is orthogonal to existing techniques. Therefore, it would be worthwhile to explore how much additional gain it can provide when combined with some of the most advanced methods, which can help assess its potential for further improving the sota in practice.

**Ethical Concerns:**

["NO or VERY MINOR ethics concerns only"]

**Final Justification:**

My main concerns have been adequately addressed during the rebuttal. I also reviewed the other reviewers' feedback. The third point raised in Reviewer sXcm's last comment—regarding performance in the very high sparsity regime—could be a limitation. Other than that, I did not notice any points that I consider to be significant issues. Therefore, I have adjusted my score accordingly.

**Limitations:**

Yes. The authors have discussed the limitations of their work.

**Quality:**

2

**Strengths And Weaknesses:**

Strengths:

1. The paper presents experiments showing evidence for the importance of learning signs in sparse training. It also shows that, in dense-to-sparse training, signs tend to stabilize early, whereas in sparse training they do not.

2. It proposes a new method that encourages the learning of signs and demonstrates through experiments that this method improves performance.


Weaknesses:

1. There has been prior work investigating the role of signs as early as 2019 [1], with findings that appear highly relevant to the observations made in this paper. There may also be more recent follow-up studies on the topic. However, the paper does not discuss any connections to these earlier works, making it unclear whether the findings—particularly those in Section 3—are novel or what they contribute beyond known results.

2. The motivation behind the proposed method lacks clarity, and the logical connection to the preceding analysis is not well established. Specifically, there appears to be a discrepancy between the problem characterization in Section 3 and the justification for the proposed approach. Section 3 highlights that, in sparse training, signs do not stabilize after the warm-up phase and continue to flip—unlike in dense-to-sparse training, where signs stabilize early. This suggests that the core issue lies in the lack of sign stability. However, the intuition provided—both in the text and in the theoretical analysis—for the proposed sign-in method is that it encourages sign flipping, as also shown empirically in Figure 4. This seems to contradict the earlier analysis, which identifies excessive sign flipping as a problem. It remains unclear how encouraging more sign flips helps address the issue of instability. There needs to be a clearer explanation or analysis of how the observations regarding sign instability is leveraged/how the proposed method connects to these observations, and why promoting sign changes would be beneficial in this context.


3. In the current experiments, it is unclear how effective the method is in advancing state-of-the-art performance or how it compares to results reported in prior work, as the only comparison is against a naive baseline. Although the method is orthogonal to existing approaches for sparse training, it would still be valuable to include comparisons with stronger baselines, or explore combinations with other methods, and conduct ablation studies to better understand its position and practical utility. This is particularly important given that the primary focus of the paper is to introduce a new method. Demonstrating its practical value more convincingly would strengthen the contribution.

[1] Zhou, Hattie, et al. "Deconstructing lottery tickets: Zeros, signs, and the supermask." Advances in neural information processing systems 32 (2019).

---

> ### Author Rebuttal · Authors · 2025-07-30
>
> We would like to express our gratitude for your time and efforts in providing valuable comments on our manuscript. Please find a detailed point-by-point response below. In case of any open questions, we would be happy to discuss them.
>
> **Rebuttal of Weaknesses**
>
> 1. We thank you for pointing us to the related work [1]. While [1] draws the insight that the signs learnt by Iterative Magnitude Pruning (IMP) and thus the initial signs are informative on small scale experiments (CIFAR10), our work is concerned with the question when signs become informative during (and the end of) training, how something similar can be achieved by PaI also in the context of larger scale experimental settings (like ImageNet). Our work is closer to [4], which analyzes how this sign learning mechanism enables LRR to outperform IMP. As mentioned, we show that this insight is more universal and holds for various mask topologies and sparsification methods. We will be happy to add this discussion.
>
> 2. As you have pointed out, dense to sparse methods flip parameter signs early in training, which stabilize during subsequent training.
>    - However, this is not the case for PaI methods as highlighted by Theorem 5.4, due to the lack of overparameterization. To mitigate this, according to Theorem 5.2, Sign-In introduces an orthogonal sign-flipping mechanism that enables useful sign flips also in the absence of overparameterization.
>    - Hence, in the case of PaI, where early sign flipping followed by sign stabilization is not possible, Sign-In induces a mechanism to learn the correct sign flips (as shown in Figure 2).
>    - Specifically, Sign-In facilitates more useful sign flips, as shown in the toy example by recovering the ground truth.
>    - To motivate this further, consider our theoretical example, where we want to train a single neuron $a \phi( w x)$, where the sign of $a \in \mathbb{R}$ is wrongly initialized. The parameter $a$ cannot be flipped due to the balancedness property. Nevertheless, all parameters in the weight vector $w \in \mathbb{R}^d$ have a chance of flipping while the activation stays non-zero for some data points. This will not necessarily bring us close to the ground truth but will lead to more sign flips (in $w$)  as in line with the general instability of the sign flipping in PaI, whereas Sign-In allows for a correct additional sign flip.
>    - Therefore we use the amount of sign flips as a proxy for the amount of additional correct sign flips on top. Further experiments with ACDC + Sign-In (Table 10) and Rigl + Sign-In (shown below) show that Sign-In performs useful sign flips which are orthogonal to the sign stabilizing mechanism in dense-to-sparse methods.
>    - So the flow of argumentation is that we first showed empirically that correct signs plus mask is enough for recovering dense performance then this is followed by a method that can provably induce a part of these correct sign flips.
>
> 3. The additional experiments below highlight the utility of Sign-In for state-of-the-art dynamic sparse training methods like RiGL and Mest. On Imagenet with a ResNet50, we see relevant improvements starting from the 90% sparsity level, where the sign optimization starts to matter more.
>
> | Method     	| s = 0.8        	| s = 0.9           | s = 0.95          |
> |----------------|---------------|-------------------|------------------|
> | RiGL       	| 75.02 ± 0.1   | 73.70 ± 0.2   	| 71.89 ± 0.07 	|
> | RiGL + Sign-In | 75.02 ± 0.1   | 74.27 ± 0.085 	| 73.07 ± 0.17 	|
>
> | Method     	| s = 0.8       | s = 0.9       | s = 0.95          |
> |----------------|---------------|-------------------|------------------|
> | Mest       	|  74.67  | 73.1  | 70.84 |
> | Mest + Sign-In |  74.74 | 73.32  | 71.41 |
>
>
> Furthermore, we show the utility of Sign-In with other PaI methods such as SNIP and Synflow in Table 7 in Appendix H. Note that [2,3] showed that the random baseline often performs better, which was the main reason to highlight it in the main part of the manuscript. We would be happy to move the additional results on SNIP and Synflow to the main part of the manuscript given an extra page.
> Additionally, we have conducted experiments with the recently proposed PaI method NPB [5], as reported below.
>
> | Model              	| s = 80%            	|s =  90%           	|s =  95%             	|
> |------------------------|-------------------|------------------|--------------------|
> | NPB C10 Res20      	| 87.30 ± 0.39  	| 85.98 ± 0.32 	| 82.06 ± 0.9 	|
> | NPB+Sign In C10 Res20  | 88.87 ± 0.016 	| 86.88 ± 0.51 	| 83.37 ± 0.65   	|
> | NPB C100 Res18     	| 74.92 ± 0.63  	| 72.79 ± 0.56 	| 71.49 ± 0.24   	|
> | NPB+Sign In C100 Res18 | 75.59 ± 0.15  	| 74.72 ± 0.18 	| 73.28 ± 0.40   	|
>
> Question
>
> We report results with additional PaI methods as well as highlight that Sign-In can improve dynamic sparse training methods like RiGL via its sign flipping mechanism. Similar improvements on sparsification with ACDC are also reported in Table 10.
>
>
> We sincerely thank you for your thoughtful feedback and constructive suggestions, which have improved our paper. We are looking forward to the discussion period and would be happy to address further questions and provide additional clarifications.
>
> [1] Zhou, Hattie, et al. "Deconstructing lottery tickets: Zeros, signs, and the supermask." NeurIPS 2019.
>
> [2] Liu, Shiwei, et al. The Unreasonable Effectiveness of Random Pruning: Return of the Most Naive Baseline for Sparse Training. ICLR 2022.
>
> [3] Gadhikar, Advait Harshal, Sohom Mukherjee, and Rebekka Burkholz. "Why Random Pruning Is All We Need to Start Sparse." ICML 2023.
>
> [4] Gadhikar, A. H., & Burkholz, R. (2024). "Masks, signs, and learning rate rewinding." ICLR 2024.
>
> [5] Pham, H., Ta, T. A., Liu, S., Xiang, L., Le, D., Wen, H., & Tran-Thanh, L. (2023). Towards data-agnostic pruning at initialization: What makes a good sparse mask? NeurIPS 2023.

---

> > ### Comment · Reviewer_G1i7 · 2025-08-03
> >
> > Thank you to the authors for the response. I think the clarification regarding the motivation makes sense, and the additional experiments have strengthened the results. I also reviewed the other reviewers' feedback and did not notice any major issues. Therefore, I have increased my score accordingly.

---

> > > ### Author Response · Authors · 2025-08-04
> > > **Response to Reviewer**
> > >
> > > We sincerely thank the reviewer for their thoughtful engagement and for taking the time to reassess our work. We are glad that our clarifications are helpful in addressing your concerns.
> > >
> > > We appreciate your constructive feedback throughout the review process, which has strengthened our paper.

---

### Official Review · Reviewer_sXcm · 2025-07-01

**Clarity:** 3
**Significance:** 2
**Originality:** 2
**Rating:** 5
**Confidence:** 4

**Summary:**

The paper studies the performance gap between pruning at initialization (PaI) methods and methods termed dense-to-sparse, in the sparse training problem domain. The authors identify sign stabilization as a distinguishing factor between the two classes of methods, where sign flips occur consistently throughout training in PaI methods while they stabilize in dense-to-sparse methods. Additionally, the authors show that the signs (identified early) alone with the final mask (fixed) is enough to recover the performance in dense-to-sparse methods, while the mask with random weights is not.

Sing-in is proposed as a method to reparameterize the fixed PaI masks to find the optimal sign assignment during training as compared to the standard training procedure. The experiments and results show that sign-in results in significant improvements in test accuracy for the PaI masks. Additionally, the authors show that the use of sign-in results in flatter minima by computing the sharpness of the loss landscape.

**Questions:**

- Can it be the case that the ability to change the connectivity and thereby starting from 0 in the newly added weights, allows the network to implicitly change or shift the signs in dynamic sparse training methods such as RiGL?

- Does sign flipping help in a very high sparsity regime? (0.98, 0.99) It may be so that sign flipping is a helpful tool when even the sparse mask is over-parameterized, but not in those high sparsity regime where mask structure may become very important.

**Ethical Concerns:**

["NO or VERY MINOR ethics concerns only"]

**Final Justification:**

I support accepting this paper based on the authors' rebuttal, additional results and their proposed changes. My previous concerns have been addressed through: clarifying the proposed method as an orthogonal tool, and adding discussion about optimal sparsity levels including discussion and ideas to identify effective sparsity thresholds. These modifications will significantly enhance the paper's clarity and highlights the contributions.

**Limitations:**

Yes

**Quality:**

3

**Strengths And Weaknesses:**

**Strengths:**

- The paper is very well written and easy to follow, while making significant contributions towards the issue of the performance gap between PaI methods and dense-to-sparse methods.

- The experiments are very well conducted, including the ones that show preliminary results of the sign stabilization, the importance of signs learned against the weight magnitudes in recovering the performance.

- The proposed method results in significant gains over the PaI methods, which is a strong result.

**Weaknesses:**

- Study of signs and their importance in lottery ticket initialization has been previously studied, which limits the contributions. However, I do note that the authors make significant contributions in terms of signs learned, the associated phases, and the proposed method itself.

**Comments:**

- Additionally, regarding “dense-to-sparse” methods. The experiments and results provided in the paper are for sparse training methods, the difference being the ability to modify or update the mask during training, albeit while using some form of dense gradients or training iterations - AC/DC being alternate, and RiGL using dense gradients. The authors should move to standard nomenclature in the literature being PaI-Pruning at Initialization and sparse training.

- There seems to be a missing link between PaI masks at initialization as compared to the result shown that signs are more important than weight magnitudes in dense-to-sparse methods in section 2. The authors should consider an experiment where PaI masks are initialized with some “good signs” to show a clear connection between signs at initialization improving performance.

- I appreciate the authors' discussion on computation and memory costs associated with sign-in. The authors should consider the following plot, x-axis being the computation budget in terms of FLOPs, and y-axis being the test accuracy, and scatter plot the results for all sparsity levels and all the methods. This gives a better picture of the budget as compared to the performance trade-off.

- The authors should consider adding the comparison with dense-to-sparse methods in the main part with dense-to-sparse+Sign-in, PaI, and PaI+Sign-in. This is a requirement since all those methods operate in a “similar” problem space where the training efficiency is paramount.

- Minor: The earlier parts of the paper including the introduction can be made much simpler, coherent and less dense. A small suggestion would be not to highlight all the results, (theoretic or empirical) as it is uneasy to follow those if the reader is already not familiar with the entire paper.

---

> ### Author Rebuttal · Authors · 2025-07-30
>
> We would like to express our gratitude for your time and efforts in providing valuable comments on our manuscript. We appreciate that you find our work well written, easy to follow and consider our results to be strong.
> Please find a detailed point-by-point response below. In case of any open questions, we would be happy to discuss them.
>
> **Rebuttal of Comments:**
> 1.  We are happy to switch to the more common terminology of the field and mainly distinguish sparse training methods and pruning at initialization.
>
> 2. By choosing the signs corresponding to warm-up or some preliminary epoch we are effectively selecting partially correct signs. To analyze the dependence on partially correct signs in more detail, we have performed additional experiments where we vary the number of correct signs for a mask identified by ACDC on a ResNet50 for ImageNet. The table below shows that fewer correct signs at initialization progressively degrade performance.
>
> | Method                     	| s=0.9   |
> |--------------------------------|---------|
> | ACDC baseline (from Table 1) 	| 74.68	|
> | ACDC Mask + 100% correct signs (from Table 1) | 74.88 |
> | ACDC Mask + 95% correct signs  |  74.44 |
> | ACDC Mask + 90% correct signs  | 73.54    |
> | ACDC Mask + 80% correct signs  | 71.81 |
> | ACDC Mask + random signs (from Table 1)  | 70.88 |
>
> 3. We would be happy to include such a figure in the main manuscript. Table 12 shows that Sign-In’s added overhead is negligible compared to the method it is applied to. Therefore, Sign-In pushes the Pareto frontier of PaI, as it achieves significantly better generalization with a similar overhead.
>
> 4. The table below presents the effect of combining Sign-In with RiGL for ResNet50 on ImageNet. We conclude that also dynamic masks can benefit from improved sign learning at higher sparsity.
>
> | Method     	| s = 0.8        	| s = 0.9           | s = 0.95          |
> |----------------|---------------|-------------------|------------------|
> | RiGL       	| 75.02 ± 0.1   | 73.70 ± 0.2   	| 71.89 ± 0.07 	|
> | RiGL + Sign-In | 75.02 ± 0.1   | 74.27 ± 0.085 	| 73.07 ± 0.17 	|
>
> 5. We would be happy to sparsify the introduction by putting less emphasis on all the individual results in the revised manuscript.
>
> Questions:
>
> 1. That is a really interesting idea. We also believe that improved sign learning could be an important mechanism contributing to the success of RiGL. To analyze this hypothesis, we have perform RiGL with two ablations in addition to re-initializing weights with 0. They are a) initializing with the weight value at pruning and b) switching the sign of the weight upon re-initalizing. We find that both perform worse than initializing at 0. This is in line with the findings of the authors of RiGL and in line with your hypothesis that initialization at 0 promotes flexibility of parameters to learn signs. (Furthermore, it perturbs the model less.) Interestingly, reinitializing with the opposite sign performs better that the original sign, suggesting that a sign flip could be helpful. We report these results for a ResNet50 on ImageNet at 90% sparsity.
>
> | RiGL ablation           | s=0.9   |
> |------------------------|---------|
> | Reinit with Zero (baseline)   	| 73.75   |
> | Reinit with Opposite Sign | 71.6  |
> | Reinit with Same Sign    | 69.51 |
>
> In addition to the mechanism of pruning and regrowing in RiGL, our results suggest that training RiGL in combination with Sign-In can lead to additional gains especially at higher sparsity (see previous section on weaknesses), as also the weights that are not rewired learn signs more effectively.
>
>
> 2. Sign-In also helps in the high sparsity regime. Inevitably, there will be a point where the connectivity between layers will break down depending on mask topology. As we see in experiments (Table 3 and 7), the performance gains are comparable for different sparsity levels. Sign-In cannot counteract the connectivity issue. However, as Sign-In is agnostic to the mask structure and the correct signs become more important for sparser masks, Sign-In should lead to similar gains in the high sparsity regime. To verify this intuition, we have conducted an additional experiment for random pruning at initialization, SNIP, and Synflow on CIFAR10 and CIFAR100. According to the table below, Sign In still outperforms the baselines. However as expected at 99% sparsity, due to mask topology instability, the differences between Sign-In and the baseline are within noise.
>
>
> | Method              | s=0.98               | s=0.99               |
> |---------------------|------------------|------------------|
> | Random c10          | 78.44 ± 0.13     | 68.56 ± 0.49     |
> | Random c10 Sign-In  | 78.78 ± 0.14     | 69.17 ± 0.29     |
> | SNIP c10            | 74.08 ± 0.72     | 65.83 ± 0.83     |
> | SNIP c10 Sign-In    | 74.22 ± 1.31     | 64.93 ± 1.13     |
> (Synflow failed)
>
> | Method                 | s=0.98               | s=0.99               |
> |------------------------|------------------|------------------|
> | Random c100            | 69.44 ± 0.47     | 67.13 ± 0.30     |
> | Random c100 Sign-In    | 70.14 ± 0.23     | 68.23 ± 0.67     |
> | SNIP c100              | 68.54 ± 0.42     | 64.74 ± 1.45     |
> | SNIP c100 Sign-In      | 69.73 ± 0.77     | 64.81 ± 0.63     |
> | Synflow c100           | 64.82 ± 0.35     | 59.57 ± 0.46     |
> | Synflow c100 Sign-In   | 65.92 ± 0.30     | 59.56 ± 0.80     |
>
>
>
> We sincerely thank you for your thoughtful feedback and constructive suggestions, which have improved our paper. We are looking forward to the discussion period and would be happy to address further questions and provide additional clarifications.

---

> > ### Comment · Reviewer_sXcm · 2025-08-03
> > **Reply to Author Response**
> >
> > I thank the authors for the detailed response and for conducting / highlighting the relevant results for the comments made. Most of my comments and questions have been addressed.
> >
> > - The major weakness in my opinion is the novelty (framing) of the approach through the study of signs in PaI, as this topic has been studied before. Signs alone are unable to fully explain or bridge the gap between PaI methods and dynamic sparse training methods or even lottery tickets. This limits the ultimate performance gains of the proposed method itself.
> >
> > - The interesting result by far is that Sign-in can help dynamic sparse training methods as well. I believe this should be highlighted and can help the framing of the paper. Sign-in can be a useful orthogonal tool for improving the performance of both PaI and dynamic sparse training methods with minimal additional costs. The emphasis can be moved from sign stabilization to learning the signs accurately, in the current version it is difficult to follow this.
> >
> > - In the rebuttal the results indicate a very high sparsity regime where sign-in does not provide any improvements over random pruning and other PaI methods. This is concerning as the trends observed in the current results is when sparsity is increased the performance gap while using sign-in vs. not using it is greater. Therefore, there exists a range of sparsity levels where sign-in can be more beneficial to both PaI and dynamic sparse training methods. This needs to be pointed out in the main part of the paper as a limitation and discussion should be added why this sparsity regime is the most beneficial. Identifying such a sparsity regime for new tasks and networks may not be possible beforehand, and this needs to be acknowledged.
> >
> > Given these concerns I maintain my original rating.

---

> ### Author Response · Authors · 2025-08-04
> **Response to Reviewer Comments**
>
> We sincerely thank you for your thoughtful feedback and for engaging in this discussion. We are happy that we were able to address most of your comments and questions. Please find our reply to your remaining concerns below.
>
> 1.  **On Parameter Signs and Our Contributions**
>
>     **1.a. Context and Novel Insights**
>
>     We would like to highlight that previous work on parameter signs by [1,2] showed that the signs learnt at the end of training with Iterative Magnitude Pruning (IMP) and Learning Rate Rewinding (LRR) contain task relevant information. In contrast, we have shown that training a good mask with a good sign initialization from scratch is in fact competitive with sparse training (in Table 1).
>
>     Our novel insights regarding signs are two-fold:
>     * **(a) Early Sign Alignment:** We show that signs are learnt early in training and important signs are already learnt as early as the 10th epoch.
>     * **(b) Universality of Sign Alignment:** We also show that this sign alignment phenomenon extends to different kinds of sparse training methods including magnitude pruning (ACDC), continuous sparsification (STR) and dynamic sparse training (RIGL). This highlights sign alignment as a more universal phenomenon.
>
>     **1.b. Theoretical Limits and Broad Applicability**
>
>     While signs largely explain the performance gap between Dynamic Sparse Training (DST) and Pruning at Initialization (PaI), our Theorem 5.4 proves that achieving the correct signs without overparameterization is a fundamental challenge for PaI. We consider identifying this limitation a strength of our work.
>
>     To address this, our method, Sign-In, introduces a sign-flipping mechanism that is orthogonal to overparameterization. As you highlighted, this allows Sign-In to improve not only PaI but also a broad range of other sparsification methods, including DST (RIGL, MEST) and ACDC.
>
> 2.  **Motivation and Versatility of Sign-In**
>
>     Our motivation to induce correct sign flips with Sign-In follows from the sign alignment observed across different sparse training methods. However, as this is not possible in PaI (as shown by Theorem 5.4), we use Sign-In to introduce an orthogonal mechanism for sign flipping as shown in Figure 2.
>
>     As you rightly noted, this mechanism's benefits extend beyond PaI. We are grateful for your positive feedback and will revise the main text to highlight this versatility, incorporating our results on RIGL, MEST, and ACDC (Table 10) to underscore our method's broad applicability.
>
> 3.  **The Relationship Between Sparsity and Performance**
>
>     We agree that Sign-In's benefit grows with sparsity, and our reasoning is as follows:
>     * **At low sparsity,** the network remains overparameterized, allowing standard optimization to be effective, so Sign-In's improvement is modest.
>     * **As sparsity increases,** the lack of overparameterization makes it difficult to flip signs. Sign-In directly alleviates this bottleneck, significantly improving performance.
>     * **At extreme sparsity,** the mask's topology itself becomes the primary constraint on expressivity, limiting the gains from any method, including ours.
>
>     We will gladly add this discussion and our supporting high-sparsity experiments to the paper.
>
> We would like to thank you again for your valuable suggestions leading to the improvement of our work.
>
> ---
> [1] Zhou, Hattie, et al. "Deconstructing lottery tickets: Zeros, signs, and the supermask." Advances in neural information processing systems 32 (2019).
>
> [2] Gadhikar, A. H., & Burkholz, R. (2024). Masks, signs, and learning rate rewinding. In The Twelfth International Conference on Learning Representations (ICLR 2024).

---

> > ### Comment · Reviewer_sXcm · 2025-08-05
> > **Response to Author Comments**
> >
> > Thank you for addressing the points I raised about the paper. The proposed modifications will improve the paper by:
> >
> > 1. Highlighting Sign-in's effectiveness for both static masks and dynamic sparse training methods
> >
> > 2. Discussing the relationship between sparsity levels and Sign-in's effectiveness
> >
> > 3. Updating the introduction section as suggested in the review.
> >
> > Based on these improvements, I have increased my score for the paper.

---

> > > ### Author Response · Authors · 2025-08-05
> > > **Response to Reviewer**
> > >
> > > Thank you again for your constructive feedback and for reassessing our work. We are happy that our clarifications were effective in addressing your concerns. Your input has been very helpful in improving our manuscript.

---

### Official Review · Reviewer_A4ar · 2025-07-02

**Clarity:** 1
**Significance:** 1
**Originality:** 1
**Rating:** 1
**Confidence:** 5

**Summary:**

This paper investigates the limitations of Pruning at Initialization (PaI) for sparse neural network training and proposes Sign-In, a novel reparameterization strategy designed to induce sign flips in parameters to improve the alignment between sparse masks and learned weights. The method introduces a dynamic reparameterization which facilitates better optimization geometry and aids in recovering correct parameter signs, especially in sparse settings. The paper supports its claims with both theoretical analysis and empirical evaluations on CIFAR-10/100, ImageNet.

**Questions:**

Please refer to the strength and weaknesses.

**Ethical Concerns:**

["NO or VERY MINOR ethics concerns only"]

**Final Justification:**

The author does not address my concern on research motivation. The necessity of the research is not strong enough. PaI is not a good start point that initialize efficient training. Therefore I don’t support acceptance of this paper.

**Limitations:**

Yes.

**Quality:**

1

**Strengths And Weaknesses:**

**Strengths**

1. The paper provides provable guarantees that Sign-In enables successful learning in previously hard-to-optimize configurations.

2. The authors evaluate their method on multiple scales and architectures across different sparsity, which demonstrate robustness.

**Weaknesses**

1. Although the author of the paper claims that the focus of the proposed method is on PaI, but the whole PaI research domain still falls in the degraded performance on model accuracy. My concern in on the motivation of the paper: does improving such method really necessary? Since the improved accuracy by Sign-In is still not comparable to many sparse training methods, the motivation is questionable.

2. The result in the paper is limited. And the existing results show limited improvements against baselines.  In the main paper, it only shows one small table of results, with 3 network structures corresponding to 3 datasets. The baseline method is Random initialization, instead of other more comprehensive PaI methods. For the baseline, it is very obvious that Random initialization has worst performance, such that the proposed method may have the best improvement performance. But if switching to a different method (e.g., SNIP), the results may not that good. Therefore, the demonstration of the results is not convincing.

3. For PaI, the author only shows SNIP, Synflow plus the proposed method. There are many other PaI method, for example, the concurrent work with SNIP, GraSP, is not compared. And many following works are also not included.

4. For Dynamic Sparse Training method like RigL, the improvement is limited (72.02->72.48). The improved accuracy is still not better than MEST (72.58) at same training budget and sparsity. I wonder if applied on even better sparse training method (PaI or DST), does the proposed method still work? This is another concern about the proposed method.

5. The motivation of the method is based on the Lottery Ticket Hypothesis, which is already proved to be falsified by many later works (also cited by the author of the paper in line 110). I may disagree with using it as the motivation or develop rigorous method based on it. The author of the paper should consider using another more rigorous theory.

6. Minor, line 37, concered -> concerned.

---

> ### Author Rebuttal · Authors · 2025-07-30
>
> We would like to express our gratitude for your time and efforts in providing valuable comments on our manuscript. Below, we elaborate on your concerns in a detailed point-by-point response. In case of any open questions, we would be happy to discuss them.
>
> **Rebuttal of Weaknesses**
>
> 1. Pruning at Initialization (PaI) is an active research field (see e.g. [1,2,7]), as solving this problem would be considered a significant breakthrough for multiple reasons:
>     - From a practical side, achieving competitive performance when training a fixed mask from scratch would enable training neural networks in low resource settings (like edge devices) - even lower than what is possible with dynamic sparse training (DST).
>     - Training models locally could also have positive implications for data privacy.
>     - Training a sparse mask would cost less memory and compute (in particular on specialized hardware such as [5]) in comparison to training denser models (or using denser gradients or larger batch sizes as in some DST approaches). The fact that the mask is fixed and known from the start in contrast to DST, would also allow for utilizing optimized kernels (or even hardware) that exploit the specific sparsity patterns to gain computational speedups during forward and backward propagation.
>     - From a conceptual side, improving PaI likely requires novel optimization approaches that provide fundamental insights into the interplay between overparameterization or sparsity and neural network optimization. Our method Sign-In, for instance, addresses the limitation of standard optimizers to effectively learn to switch parameter signs. In overcoming this limitation, it provably solves an optimization problem that is hard in the standard setting.
>
> 2. We would like to highlight Table 7 in the appendix, which presents results for Sign-In applied to SNIP and Synflow alongside random masks. In addition to these PaI methods, we also report new results for NPB (a recent PaI method) and RiGL (a dynamic sparse training method) below.
>
> 3. Many other methods do not outperform random pruning at initialization (PaI) as shown in [3,4]. For that reason, we focused on demonstrating the effectiveness of Sign-In on random pruning in the main manuscript. With an extra page of space, we would be happy to move our results on SNIP and Synflow to the main part of the paper.
> In addition, as suggested by the reviewer, we perform experiments on PaI with the NPB method [7], results shown below on both CIFAR10 with a ResNet20 and CIFAR100 with a ResNet18:
>
> | Model              	| s = 0.8            	|s =  0.9           | s =  0.95             	|
> |------------------------|-------------------|------------------|--------------------|
> | NPB C10 Res20      	| 87.30 ± 0.39  	| 85.98 ± 0.32 	| 82.06 ± 0.9 	|
> | NPB+Sign-In C10 Res20  | 88.87 ± 0.016 	| 86.88 ± 0.51 	| 83.37 ± 0.65   	|
> | NPB C100 Res18     	| 74.92 ± 0.63  	| 72.79 ± 0.56 	| 71.49 ± 0.24   	|
> | NPB+Sign-In C100 Res18 | 75.59 ± 0.15  	| 74.72 ± 0.18 	| 73.28 ± 0.40   	|
>
>
> 4. We assume your comment refers to Table 8 in the appendix.
>     - The purpose of Table 8, is to investigate how Sign-In performs for different fixed mask topologies that are derived from running different sparse training methods. Each method in Table 8 is initialized with a fixed mask which has been identified by a specified pruning method and then trained from scratch. The experiment is designed like this to assess how close we come to solving PaI (where only the mask is given). Matching a target performance (obtained by the sparse training method) would mean solving PaI. Hence, the numbers in Table 7 highlight this gap between PaI and other sparse training methods like RiGL or ACDC. The actual performances for these methods are reported in Table 1 for comparison (which for RiGL is 73.75%).
>     - As per your comment, we can still ask the question if our method Sign-In could improve dynamic sparse training. To this end, the table below for ResNet50 on ImageNet shows that Sign-In can boost RiGL especially at high sparsity.
>     - It can also improve Mest, as we demonstrate with a ResNet50 trained on ImageNet.
>
> | Method     	| s = 0.8        	| s = 0.9           | s = 0.95          |
> |----------------|---------------|-------------------|------------------|
> | RiGL       	| 75.02 ± 0.1   | 73.70 ± 0.2   	| 71.89 ± 0.07 	|
> | RiGL + Sign-In | 75.02 ± 0.1   | 74.27 ± 0.08 	| 73.07 ± 0.17 	|
>
>
> | Method     	| s = 0.8       | s = 0.9       | s = 0.95          |
> |----------------|---------------|-------------------|------------|
> | Mest       	|  74.67  | 73.1  | 70.84 |
> | Mest + Sign-In |  74.74 | 73.32  | 71.41 |
>
>
> 5. Neither the premise of our work nor our theory is based on the lottery ticket hypothesis (LTH).
>     - We cite the LTH as an important line of work that highlights the difficulty of training sparse neural networks from scratch. The fact that lottery tickets might not exist or are more difficult to find in large-scale settings supports our point. [6] hypothesizes that the main challenge for lottery tickets is in fact to learn parameter sign flips. It argues that learning rate rewinding (LRR), which gives up on the lottery ticket idea, succeeds by improved sign identification. Table 1 demonstrates that the superior sign learning ability is actually a more universal property of contemporary sparsification and DST methods, which do not find lottery tickets but utilize different forms of overparameterization.
>     - Our proposed parameterization Sign-In further boosts sign learning. Our theory even proves this in the setting of [6].
> Another way to frame our innovations could be to cast them as exploration of controllable Riemannian gradient descent for better generalization. While we demonstrate its utility for PaI, the tools can also be used for different purposes.
>
> 6. We will correct the typo in the revised manuscript. Thank you for pointing it out.
>
> In summary, we have argued why PaI is a worthwhile endeavour to enable computing on edge devices and gain theoretical insights into the role of overparameterization in (sparse) deep learning.
> Moreover, we have referred to more results with other PaI methods (Sign-In applied to SNIP and Synflow in Table 7) and added experiments with NPB and the dynamic sparse training (DST) method RiGL, finding that our method Sign-In leads to consistent improvements.
> Furthermore, we have clarified our theoretical foundation and motivated it from an optimization angle.
>
> We sincerely thank you for your thoughtful feedback and constructive suggestions, which have improved our paper. We are looking forward to the discussion period and would be happy to address further questions and provide additional clarifications upon request.
>
>
> [1] Iurada, Leonardo, Marco Ciccone, and Tatiana Tommasi. "Finding lottery tickets in vision models via data-driven spectral foresight pruning." CVPR 2024.
>
> [2] Adnan, Mohammed et al. “Sparse Training from Random Initialization: Aligning Lottery Ticket Masks using Weight Symmetry.” ICML 2025
>
> [3] Liu, Shiwei, et al. The Unreasonable Effectiveness of Random Pruning: Return of the Most Naive Baseline for Sparse Training. ICLR 2022
>
> [4] Gadhikar, Advait Harshal, Sohom Mukherjee, and Rebekka Burkholz. "Why Random Pruning Is All We Need to Start Sparse." ICML 2023.
>
> [5] Thangarasa, Vithursan, et al. "Sparse-IFT: Sparse Iso-FLOP Transformations for Maximizing Training Efficiency." International Conference on Machine Learning. PMLR, 2024.
>
> [6] Gadhikar, A. H., & Burkholz, R. (2024). Masks, signs, and learning rate rewinding. ICLR 2024.
>
> [7] Pham, H., Ta, T. A., Liu, S., Xiang, L., Le, D., Wen, H., & Tran-Thanh, L. Towards data-agnostic pruning at initialization: What makes a good sparse mask?  NeurIPS 2023

---

> > ### Comment · Reviewer_A4ar · 2025-08-03
> > **Thank you for your rebuttal**
> >
> > I appreciate the author's effort in the rebuttal. The author provides feedbacks to many of my questions and concerns, but there still some followup questions/concerns:
> >
> > For the paper's motivation, I agree with most of the statements in author's feedback, that the sparse training can promotes practical application, more privacy, more efficient, etc. However, what I am concerned most is the accuracy of PaI. What I mean in my initial review is that since the accuracy of PaI is far less comparable to the other sparse training method such as DST, then what is the point to improve it using the proposed method in the paper. Although Sign-In can improve accuracy, but it still not as accurate as DST. For example, on ResNet-50 ImageNet with 90% sparsity, MEST achieves 72.58% according to their papers, while the Sign-In with 90% sparsity achieves 72.19% using same training budget. This demonstrate that the PaI method is already less comparable to DST although they are all sparse training approaches. Why not focusing on DST since it achieves better accuracy in learning. Considering MEST is a year 2021 paper (which is relatively old), I would say that improving PaI method is not necessary.
> >
> > Another reason is that DST method has great potential for accuracy boost when scale the training budget. The RigL and MSET paper all demonstrate significant accuracy improvement (e.g., 72.58% ->76.13% in MEST at 90% sparsity with 1.7x training budget) when they scaling up the training budget. However, for a static sparse masking method such as PaI, it is not possible. It seems that DST has more potential than PaI, and that's another reason that I question the motivation of the paper.
> >
> > About the memory and computation cost in static sparse training with PaI and DST, MEST is also Memory-Economic Sparse Training, and I believe they also maintains full sparse for their computation although they change their sparse mask. Therefore it is not completely true that PaI is more memory and computation efficient. And for the batch size setting, it is common techniques to accumulate gradient with small batch size to achieve effect of large batch size training. And for optimized hardware kernel to support DST mechanism, I believe that if a method is superior in its performance, the optimization on the hardware is necessary. That's why NVIDIA support 2:4 sparse pattern in their Ampere architecture.
> >
> > Since the theory does not related to lottery ticket, I suggest the author remove the term lottery ticket in the paper to avoid confusion, since the LTH itself is a misleading work.

---

> ### Author Response · Authors · 2025-08-03
> **Reply to Reviewer response**
>
> We sincerely thank you for your thoughtful feedback and for engaging in this discussion. We appreciate your comments that allow us to clarify the contributions and context of our work.
>
> Below, we highlight a) why PaI is an important research direction and how it creates a hard test bed for new training algorithms that can benefit sparse training in general. b) Note that our proposal, Sign-In, does not only benefit PaI but also improves other sparse training methods, including DST.
>
> * **Sign-In Improves Existing DST Methods:** Our work is not limited to Pruning at Initialization (PaI). New results show that **Sign-In** also enhances contemporary Dynamic Sparse Training (DST) methods:
>     * At 90% sparsity: RIGL improves from 73.7% to **74.27%**.
>     * At 90% sparsity: MEST improves from 73.1% to **73.32%**.
>     * These results are consistent with our findings for ACDC (Appendix, Table 10), demonstrating the broad benefit of improving sign learning.
>
> * **PaI Remains a Critical Research Area:** While we agree DST currently achieves higher accuracy, PaI's core constraint—a **fixed sparse mask**—makes it a unique and valuable research problem. It is a different training principle that is useful in specific very low resource settings. Note that altering the mask during training introduces additional overhead and is **not free of cost**.
>     * **Challenge:** The fixed mask makes optimization fundamentally harder, requiring novel techniques to succeed with less overparameterization (as corroborated by our Theorem 5.4).
>     * **Opportunity:** A static mask enables significant computational speedups via specialized kernels [1], an advantage that is less pronounced for dynamic masks in DST.
>
> * **Our Core Contribution:** We identify the importance of sign alignment in sparse training and propose **Sign-In**, a simple and effective reparameterization to improve performance by introducing useful sign flips in the absence of overparameterization. As shown above, its benefits are applicable across different sparse training paradigms.
>
> * **Context from the Lottery Ticket Hypothesis (LTH):** The LTH established the core challenge in sparse training: successfully training a sparse mask.
>     * It demonstrated that standard optimizers require special initializations that are practically unattainable without first training an overparameterized model.
>     * Our work builds on this by explaining **why** it is hard to optimize in the absence of overparameterization and, based on that insight, proposing an improved training approach.
>
> * **Proposed Action:** To reflect this broader impact, we will move the results for ACDC, RIGL, and MEST into the main body of the paper.
>
> We thank you again for helping us strengthen our work.
>
> [1] Thangarasa, Vithursan, et al. "Sparse-IFT: Sparse Iso-FLOP Transformations for Maximizing Training Efficiency." International Conference on Machine Learning. PMLR, 2024.

---

> > ### Author Response · Authors · 2025-08-06
> > **Request for Reviewer Response**
> >
> > We thank you for your time and effort in reviewing our paper. As the author–reviewer discussion deadline is fast approaching, we kindly request your response to our rebuttal at your earliest convenience. We would be happy to answer any open questions upon request.

---

### Note · Authors · 2025-08-11

We sincerely thank all reviewers for their diligent, insightful feedback and constructive critiques, which have helped us sharpen our arguments and clarify our core contributions.

In the following, we provide a summary of our rebuttal and key contributions, incorporating feedback from all reviewers.

### Contributions

* **Early Sign Alignment:** We show that signs are learned very early in training with high degrees of overparameterization.
* **Universality of Sign Alignment:** This phenomenon extends to various sparse training methods, including magnitude pruning (ACDC), continuous sparsification (STR), and dynamic sparse training (RiGL). This highlights sign alignment as a more universal phenomenon.
* **Sign-In Parameterization:** We propose a neural network re-parameterization, Sign-In, that improves sparse training by promoting sign learning.
* **Provably Inducing Sign Alignment:** Using tools from Riemannian gradient flows and dynamical systems, we prove in a toy setup that Sign-In induces correct sign flips in hard cases in which standard gradient flow would fail.

### Addressing Specific Feedback

1.  **PaI Remains a Critical Research Area:** PaI's fixed-mask constraint is a unique and valuable challenge. This principle is useful in very low-resource settings, as altering the mask during training is not without cost.
    * **Challenge:** The fixed mask makes optimization fundamentally harder, requiring novel techniques to succeed with less overparameterization (as corroborated by our Theorem 5.4), which in turn turn out to be useful in more general sparse training (see Point 2).
    * **Opportunity:** A static mask enables significant computational speedups via specialized kernels, an advantage that is less pronounced for dynamic masks in DST.
2.  **Sign-In Improves Existing DST Methods:** Our work is not limited to Pruning at Initialization (PaI). New results show that Sign-In also enhances contemporary Dynamic Sparse Training (DST) methods.
    * At 90% sparsity: RiGL improves from 73.7% to **74.27%**.
    * At 90% sparsity: MEST improves from 73.1% to **73.32%**.
    * These results are consistent with our findings for ACDC (Table 10), demonstrating the broad benefit of improving sign learning.

We believe these points clarify the significance and broad applicability of our work, showing its value both as a novel contribution to sparse training and as a tool for improving existing methods.

Thank you once again for your time and efforts.

---

### Decision · Program_Chairs · 2025-09-17

**Decision:**

Accept (poster)

**Comment:**

The authors explore the role of signs in both sparse training and pruning at initialization (PaI), and propose a method (Sign-In) to induce sign flips in parameters to improve the alignment of masks for sparse training. While there is previous literature exploring signs and mask alignment in the context of sparse training, the authors explore a different context than previous work and provide analysis on PaI and Dynamic Sparse Training (DST) methods.

The reviewers outlined the strengths of the work in the experimental evaluation, the improved performance of Sign-In over existing PaI baselines, and the strong theoretical backing for the method. While the reviewers did initially note several important weaknesses in the work, the discussion and rebuttal addressed almost all of the reviewers' concerns, with the exception perhaps of those of Reviewer A4ar. I would like to thank all the reviewers and the authors for their productive and extensive discussions, and their genuine engagement, during the rebuttal period which it appears to have improved the work substantially, which is the best outcome one can hope for from peer review.

It appears Reviewer A4ar's main issue with the work that remains unaddressed through the rebuttal is with the potential impact of the PaI research field itself in the context of much stronger sparse training (e.g. DST) baselines, and by extension, a lack of impact and audience for the author's work. The potential impact of PaI is of course debatable, and Reviewer A4ar's skepticism of PaI in general is no doubt shared amongst many within the research community (perhaps even myself). However, regardless of where one stands on PaI, I believe it is clear that there is a significant audience within the ML research community interested in PaI, and this work shows significant improvements on existing PaI baselines which will be of interest to many. Moreover, I believe it's important to point out that the authors demonstrate insights and improvements to DST/sparse training methods including RiGL and MEST, showing the work has a much broader reach and potential impact than PaI alone.